# KAT3-dependent acetylation of cell type-specific genes maintains neuronal identity in the adult mouse brain

Michal Lipinski [1,3], Rafael Muñoz-Viana [1,3], Beatriz del Blanco[1,3], Angel Marquez-Galera[1], Juan Medrano-Relinque[1], José M. Caramés[1], Andrzej A. Szczepankiewicz [2], Jordi Fernandez-Albert[1], Carmen M. Navarrón [1], Roman Olivares[1], Grzegorz M. Wilczyński[2], Santiago Canals [1], Jose P. Lopez-Atalaya[1] & Angel Barco [1✉]

The lysine acetyltransferases type 3 (KAT3) family members CBP and p300 are important transcriptional co-activators, but their specific functions in adult post-mitotic neurons remain unclear. Here, we show that the combined elimination of both proteins in forebrain excitatory neurons of adult mice resulted in a rapidly progressing neurological phenotype associated with severe ataxia, dendritic retraction and reduced electrical activity. At the molecular level, we observed the downregulation of neuronal genes, as well as decreased H3K27 acetylation and pro-neural transcription factor binding at the promoters and enhancers of canonical neuronal genes. The combined deletion of CBP and p300 in hippocampal neurons resulted in the rapid loss of neuronal molecular identity without de- or transdifferentiation. Restoring CBP expression or lysine acetylation rescued neuronal-specific transcription in cultured neurons. Together, these experiments show that KAT3 proteins maintain the excitatory neuron identity through the regulation of histone acetylation at cell type-specific promoter and enhancer regions.

[1] Instituto de Neurociencias, Universidad Miguel Hernández–Consejo Superior de Investigaciones Científicas, Avenida Santiago Ramón y Cajal, s/n, Sant Joan d'Alacant, 03550 Alicante, Spain. [2] Nencki Institute of Experimental Biology, Polish Academy of Science, 3 Pasteur Street, 02-093 Warsaw, Poland. [3]These authors contributed equally: Michal Lipinski, Rafael Muñoz-Viana, and Beatriz del Blanco. ✉email: abarco@umh.es

Neuronal identity is established through the action of a particular class of transcription factors (TFs), referred to as terminal selectors, that bind to cis-regulatory elements of terminal identity genes and regulate the establishment of neuron-specific gene programs[1,2]. Many of these terminal selectors are also required at later stages of development[3], which led to the idea that the identity of postmitotic neurons needs to be actively maintained throughout life by the action of the same TFs that controlled the last steps of differentiation[4]. However, the mechanisms responsible for such maintenance remain elusive. Chromatin-modifying enzymes are also involved in identity acquisition[5] and maintenance[6].

The KAT3 family of transcriptional co-activators comprises the CREB binding protein CBP (aka KAT3a) and the E1A binding protein p300 (aka KAT3b)[7]. Both proteins have a lysine acetyltransferase (KAT) catalytic domain and interact with numerous other proteins, including histones, TFs, the RNA polymerase II complex (RNAPII), protein kinases and other chromatin-modifying enzymes[8] that also are substrates of the CBP/p300 catalytic activity[9]. Consistent with this central function, KAT3 proteins play a critical and dose-dependent role during neurodevelopment[10]. Deletions and inactivating mutations in KAT3 genes cause early embryonic death and neuronal tube closure defects[11], whereas hemizygous mutations cause a syndromic disorder associated with intellectual disability known as Rubinstein–Taybi syndrome (RSTS)[12]. After birth, CBP loss in postmitotic forebrain neurons impairs memory in specific tasks, but does not interfere with cell viability and animal survival[13–15]. The consequences of p300 loss in adult neurons have been less explored, but the only study conducted so far revealed very mild defects[16]. The relatively modest consequences of the post-embryonic loss of either one of these central transcriptional regulators could indicate that their role is limited to development. Alternatively, since both proteins are still expressed in adult neurons, they could be functionally redundant and thus mutually compensate for each other loss.

To define KAT3-specific functions in the adult brain, we generate mice with inducible and restricted ablation of CBP and/or p300 in forebrain principal neurons. The characterization of these strains reveals that while both proteins are individually dispensable for the normal function of mature neurons, their combined ablation has devastating and rapid consequences: dendrites retract, synapses are lost, and electrical activity and neuron-specific gene programs established during development are shutdown. These phenotypes lead to severe neurological defects and premature death. Epigenomic screens and rescue experiments demonstrate that both KAT3 proteins are jointly essential for maintaining the identity of excitatory forebrain neurons (and likely other cell types) throughout life by preserving acetylation levels at cell type-specific genes and enhancers.

## Results

**Combined loss of CBP and p300 in adult excitatory neurons**. To elucidate the neuronal roles of CBP and p300 in the adult brain, we selectively eliminated CBP, p300, or both KAT3 proteins in forebrain excitatory neurons of adult mice using the inducible Cre-recombinase driver CaMKIIα-CreERT2 (Fig. 1a)[17]. The three inducible forebrain-specific knockout strains (referred to as CBP-ifKO, p300-ifKO, and dKAT3-ifKO, respectively) do not show any neurological symptom before gene(s) ablation (Supplementary Table 1). After tamoxifen (TMX) treatment of 3-month old mice, loss of immunoreactivity was observed in virtually 100% of the pyramidal neurons in the CA1 and cortex, and granule neurons in the dentate gyrus (DG; Fig. 1b and Supplementary Fig. 1a), while brain regions in which the CaMKIIα

promoter is not active, such as the cerebellum and the basal ganglia, were spared[17]. Importantly, the loss of either one of these paralogous proteins did not affect the expression level of the other (Supplementary Fig. 1b).

Consistent with the characterization of other forebrain-specific KO strains for CBP or p300[13,14,16], CBP- and p300-ifKO mice had a normal lifespan and showed no overt neurological abnormalities (Fig. 1c). However, the situation was markedly different in dKAT3-ifKOs. When CBP and p300 were simultaneously removed in the forebrain of adult mice, the animals displayed a dramatic and rapidly progressing deteriorating phenotype (Supplementary Table 1 and Supplementary Fig. 1c, d). In the 1st week following TMX administration, the mice were hyperactive and frequently froze in bizarre positions during manipulation. One week later, the same mice showed severe ataxia, and loss of the righting reflex, escaping response and tail-suspension-evoked stretching (Supplementary Movie 1). Although different animals reached the terminal phenotype at different times, initially all the mice died within the first 2–3 weeks after TMX administration. Upon facilitating access to food and water, survival increased by about 40% within the 1st month after TMX (Fig. 1c), which suggests that the ataxic mice had problems reaching the cage feeder and bottle. Remarkably, mice carrying just a single functional allele encoding either one of the two KAT3s did not die prematurely and showed normal basic reflexes (Supplementary Table 1 and Supplementary Fig. 1d, e).

**Synaptic loss and reduced electrical activity in dKAT3-ifKOs.** Consistent with the severe neurological phenotype, the very few dKAT3-ifKOs that survived for more than 2 months after TMX (less than 5%) showed a reduction of cerebral cortex volume, whereas other brain regions not targeted by gene deletion, such as the cerebellum, were unaffected (Fig. 1d). Interestingly, histological analyses indicated that this reduction was caused by a loss of neuropils rather than a loss of neuronal cells. For instance, in the hippocampus of dKAT3-ifKOs, the *stratum pyramidale*—a layer occupied by the somas of CA1 pyramidal neurons—had a normal thickness and did not show any obvious sign of neurodegeneration (Fig. 1e and Supplementary Fig. 2a, b) even several weeks after TMX treatment. In contrast, the *stratum radiatum*—a layer occupied by the basal dendrites of the CA1 neurons—was markedly thinner in dKAT3-ifKOs as soon as 1 month after TMX (Fig. 1e). In agreement with these quantifications, Golgi staining revealed a retraction of dendrites in the DG (Fig. 1f, g) and electron microscopy analyses confirmed the massive loss of synapses in dKAT3-ifKOs (Fig. 1h, i). Furthermore, as early as 2 weeks after gene ablation, dKAT3-ifKOs implanted with multichannel electrodes displayed a dramatic reduction in both spontaneous (Fig. 1j, k and Supplementary Fig. 2c, d) and evoked (Fig. 1l, m and Supplementary Fig. 2e) electrical activity. Overall, these data demonstrate that the simultaneous loss of CBP and p300 alters neuronal morphology and impairs electrical responses leading to dramatic neurological defects.

Strikingly, these severe changes occur in the absence of cell death. TUNEL staining (Supplementary Fig. 3a) and immunostaining against active caspase 3 (Fig. 1n), which label cells undergoing apoptosis, were both negative. Immunostaining against the histone H2A.X variant (Supplementary Fig. 3b) that labels cells suffering DNA damage was also negative. EM images revealed a *stratum pyramidale* with a similar density of somas in dKAT3-ifKOs and control littermates (Supplementary Fig. 3c). Neuronal nuclei did not present apoptotic bodies, although the nucleoplasm appeared clearer and with slightly larger heterochromatic domains in dKAT3-ifKOs than in

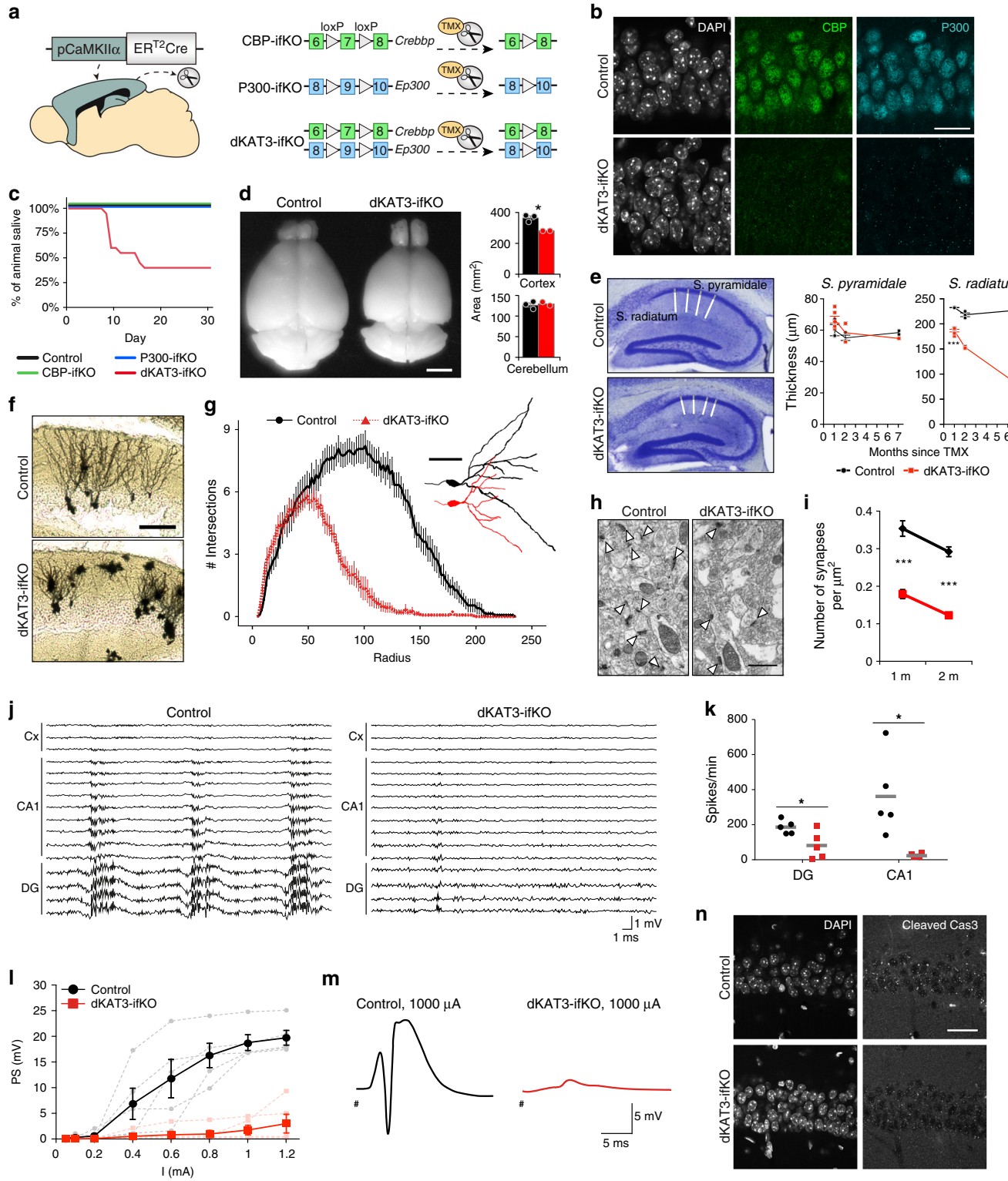

controls (Supplementary Fig. 3d). To monitor the evolution of double KO neurons in a cell autonomous manner, we infected the DG of the $Crebbp^{f/f}::Ep300^{f/f}$ (dKAT3-floxed) mice with adeno-associated virus (AAV) expressing Cre recombinase under the synapsin promoter (Supplementary Fig. 3e). Immunostaining confirmed the efficient and complete elimination of CBP and p300 in granule neurons in the absence of detectable neurodegeneration even 10 weeks after genes ablation (Supplementary Fig. 3f, g).

**Maintenance of neuronal identity requires at least one KAT3.** To determine the molecular basis of the abovementioned phenotypes, we conducted a RNA-seq screen in the hippocampus of dKAT3-ifKOs and control littermates. Differential gene expression profiling revealed 1952 differentially expressed genes (DEGs; $|log2FC| > 1$) in dKAT3-ifKOs, with a clear preponderance both in number and magnitude of gene downregulations (Fig. 2a, b, Supplementary Fig. 4a, and Supplementary Data 1). Gene Ontology (GO) enrichment analysis indicated that

**Fig. 1 Loss of both KAT3 proteins causes severe neurological alterations. a** Genetic strategy for the production of inducible, forebrain-specific CBP, p300, and double KAT3 knockouts. **b** Double immunostaining against CBP and p300 in the CA1 region. The analysis was performed twice with different sets of animals. Scale: 10 µm. **c** Survival of the three ifKO lines after TMX administration (control, $n = 14$; dKAT3-ifKO, $n = 20$). **d** Left: representative images of control and dKAT3-ifKO brains 2 months after TMX. Scale: 5 mm. Right: quantification of cortical and cerebellar sizes (control, black bars, $n = 3$; dKAT3-ifKO, red bars, $n = 2$). Two-tailed $t$-test: *$p$-value = 0.024. **e** Left: Nissl staining of hippocampi 2 months after TMX. Right: quantification of the thickness of the strata pyramidale and radiatum of control (1 m, $n = 3$; 2 m, $n = 3$; 7 m, $n = 2$) and dKAT3-ifKO mice (1 m, $n = 6$; 2 m, $n = 2$; 7 m, $n = 1$) at different time points after TMX. Points show separate observations. Two-way ANOVA; st. radiatum: ***$p$-val$_{genot}$ = 1.09e−09, $p$-val$_{time}$ = 2.26e−05, **$p$-val$_{genot:time}$ = 0.006; st.pyramidale: *$p$-val$_{genot}$ = 0.036. Post-hoc Tukey HSD comparison st. radiatum 1 month control—1 month dKAT3-ifKO; ***$p$-value = 1.7e−06. **f** Representative images of Golgi staining in the dentate gyrus 1 month after TMX ($n = 4$). Scale: 100 µm. **g** Sholl analysis of DG neurons 1 month after TMX (32 neurons from 4 control mice; 38 neurons from 4 dKAT3-ifKOs). The right inset shows representative Neurolucida-reconstructed neurons. Scale: 50 µm. **h** Electron microscopy images showing the stratum radiatum 1 month after TMX. Arrowheads indicate positions of the synapses ($n = 3$). Scale: 1 µm. **i** Number of synapses per µm$^2$ in the stratum radiatum (average of 30 areas from 3 mice per condition). Two-tailed $t$-test; ***$p$-val$_{1 month}$ < 0.0001; **$p$-val$_{2 months}$ < 0.001. **j** Representative in vivo electrophysiological recordings of spontaneous activity across cortical and hippocampal layers. **k** Firing frequency (spikes per minute) recorded in CA1 and DG. SUA: single unit analysis ($n = 5$). Gray lines represent the mean values. Two-tailed $t$-test: *$p$-val$_{CA1}$ = 0.022, $p$-val$_{DG}$ = 0.028. **l** Population spike (PS) amplitude in DG after the application of increasing intensities in the perforant pathway ($n = 5$). Lighter-colored elements show the results for each animal separately. Two-tailed $t$-test. **m** Representative evoked potentials waveforms (PS) to 1 mA stimulation. **n** Immunohistochemistry for cleaved Cas3 shows no sign of apoptosis in the CA1 subfield 1 month after TMX. The analysis was performed twice with different sets of animals. Scale: 30 µm. In panels (**d**), (**e**), (**g**), (**i**), and (**l**), data are presented as mean values ± SEM. Source data for graphs in panels (**c**), (**d**), (**e**), (**g**), (**l**), (**k**), and (**l**) are provided as a Source data file.

these downregulations affect a large number of neuronal functions (Fig. 2c, blue bars). Hundreds of genes with neuronal functions such as genes encoding channels and proteins important for synaptic transmission were severely downregulated in the dKAT3-ifKO hippocampus, which explains the reduced neuronal firing and lack of electrical responses. Gene upregulation was much more restricted, including a modest inflammatory signature (Fig. 2c, red bars) but no activation of cell death pathways (Supplementary Fig. 4b). In fact, several positive regulators of neuronal death were strongly downregulated in dKAT3-ifKOs (e.g., *Hrk*; Supplementary Fig. 4c). Consistent with the survival of these cells, housekeeping genes remained largely unchanged (Fig. 2b and Supplementary Fig. 4d). Immunodetection experiments for neuronal proteins like CaMKIV, NeuN, and hippocalcin confirmed the dramatic loss of expression of neuronal proteins (Fig. 2d, e and Supplementary Fig. 4e). Notably, the loss of neuronal markers expression was not detected in mice bearing a single functional KAT3 allele (Supplementary Fig. 4f), indicating that this minimal gene dose is sufficient to preserve their expression.

To determine if these deficits were cell-autonomous, we examined the hippocampus of dKAT3-floxed mice monolaterally infected with Cre-recombinase-expressing AAVs. We observed a dramatic loss of neuronal marker expression only in the transduced DG (Fig. 2f). Moreover, the cells depleted of KAT3 proteins completely failed to respond to kainic acid, a strong agonist of glutamate receptors, further confirming the loss of excitatory neuron properties (Supplementary Fig. 5). Similarly, experiments in primary neuronal cultures (PNCs) produced from the hippocampi of *Crebbp^{f/f}::Ep300^{f/f}* embryos and infected with a Cre-recombinase-expressing lentivirus (Fig. 2g) did not show enhanced neuronal death after the simultaneous elimination of CBP and p300 (Supplementary Fig. 6a, b). These cells, however, showed abnormal morphology (Fig. 2h), downregulation of neuron-specific transcripts and proteins (Fig. 2i and Supplementary Fig. 6c), and severe hypoacetylation of KAT3 targets (Supplementary Fig. 6d).

**dKAT3-KO neurons acquire a novel, molecularly undefined fate.** The altered morphology, electrophysiological properties, and gene expression all suggest that the excitatory neurons rapidly lose their identity after the elimination of both KAT3 genes. To tackle this hypothesis, we compared the set of DEGs in dKAT3-ifKOs with transcriptome information for the different cell types in the adult mouse brain using single-cell RNA-seq (scRNA-seq) data from the mouse cortex[18] (Fig. 3a). Our analysis revealed that the genes typically expressed in CA1 and S1 pyramidal neurons were significantly downregulated in the hippocampus of dKAT3-ifKOs, whereas other cell-type-specific transcriptional programs were unaffected except for a modest increase of microglia genes related to inflammation. Importantly, although identity loss is often associated with dedifferentiation (i.e., the regression to an earlier stage of differentiation), we did not detect an upregulation of stemness genes[19], nor neuronal stem cell (NSC)- or neuroprogenitor (NPC)-specific gene expression[20] (Fig. 3a and Supplementary Fig. 7a, b). Trans-differentiation can be also discarded because we did not detect the upregulation of the transcriptional signatures of other brain cell-types.

To determine more precisely the fate of excitatory neurons after losing their identity, we conducted single-nucleus RNA-seq analyses 2 and 5 weeks after TMX treatment (Fig. 3b, c and Supplementary Fig. 7c). We observed a progressive confluence of CA1/CA3 pyramidal neurons and DG granule neurons in a common cell cluster depleted of neuronal type-specific markers (Fig. 3d, Supplementary Fig. 7d, e, f, g and Supplementary Data 2) and in which no other distinctive markers appear (Fig. 3e, Supplementary Fig. 7h, i, and Supplementary Data 2). Of note, these cells do not show a reduction in RNA content and still express a large number of transcripts, confirming that the reduction is restricted to a specific gene set (Supplementary Fig. 8). To further investigate the transitional states of dKAT3-KOs neurons, we used manifold learning leveraged in nearest-neighbor information to automatically organize excitatory neurons in trajectories along a principal tree[21]. This analysis produced a topological structure with five main branches and three bifurcation points (Fig. 3f). Pseudotime analysis revealed a trajectory in which cells travel from the roots in branches A (corresponding to DG granule cells), B and C (both populated by pyramidal neurons) towards branches D and E that are almost exclusively populated by the cells depleted of cell-type-specific markers (Fig. 3g). Altogether, these results indicate that the different main types of excitatory hippocampal neurons in dKAT3-KOs lose their neuronal identity, but do not die, dedifferentiate or transdifferentiate to other cell types. Instead, these cells seem to be trapped in an undifferentiated, non-functional interstate deadlock.

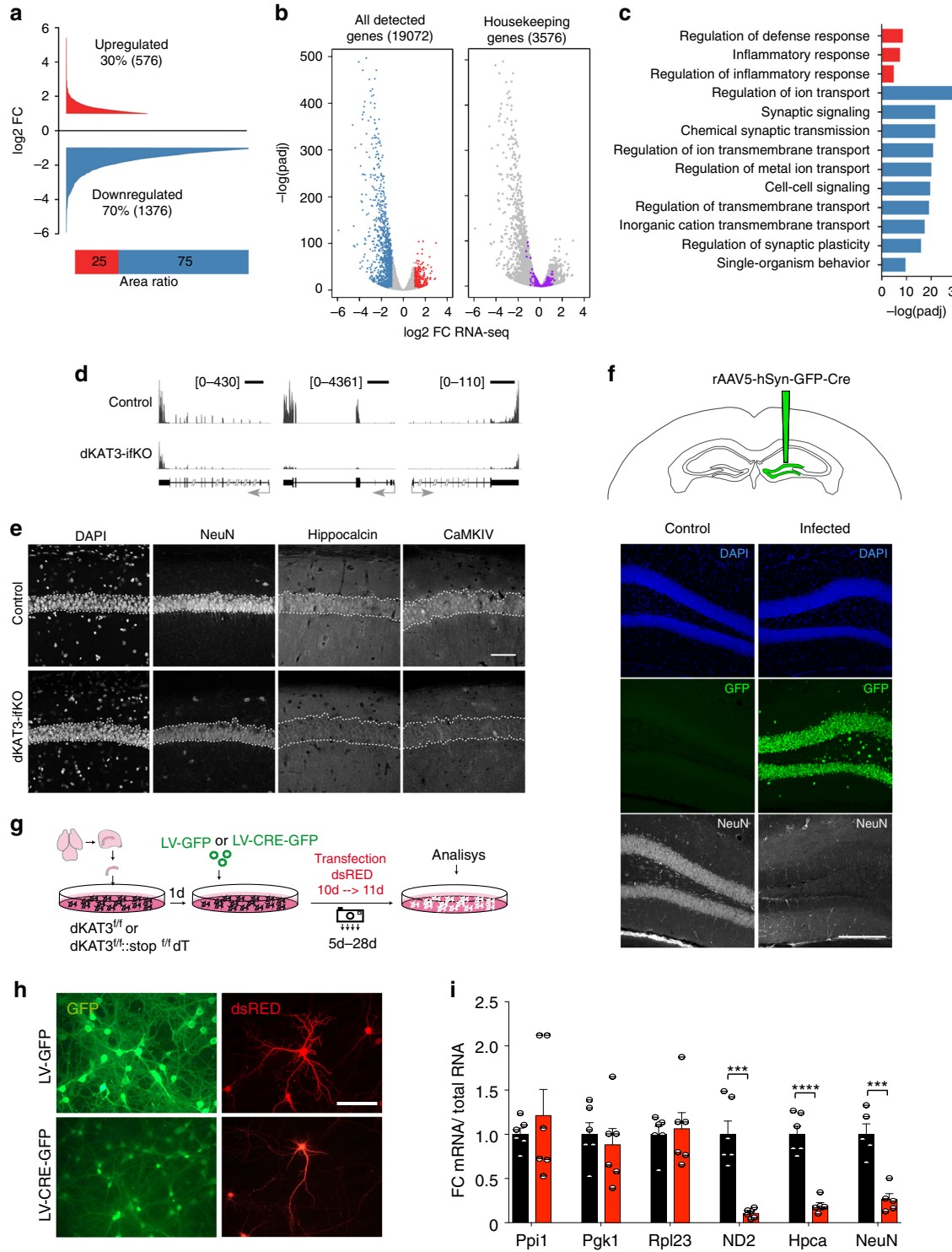

**CBP and p300 bind to the same regulatory regions**. The evidence above indicates that KAT3 proteins play a redundant role in preserving neuronal identity. To explore the basis of such redundancy, we next mapped the occupancy of hippocampal chromatin by CBP and p300. Chromatin immunoprecipitation followed by whole-genome sequencing (ChIP-seq) using antibodies that differentiate between the two paralogous proteins (Supplementary Figs. 1b and 9a) retrieved 37,359 peaks in the chromatin of wild type mice (Fig. 4a, Supplementary Fig. 9b, and Supplementary Data 3). Considering the different efficiencies of the two antibodies, we detected an almost complete overlap

between the CBP and p300 peaks throughout the whole genome (Fig. 4b and Supplementary Fig. 9c). This result indicates that these two essential epigenetic enzymes largely occupy the same sites in neuronal chromatin and underscores their functional redundancy.

As both proteins are ubiquitously expressed yet gene ablation only takes place in excitatory neurons, the signal detected in the hippocampal chromatin of dKAT3-ifKOs must correspond to CBP/p300 binding in other cell types. We combined this information with ATAC-seq (a technique that investigates chromatin occupancy and requires much lower input than

**Fig. 2 Hippocampal cells lacking KAT3 fail to express neuronal-specific genes. a** Cumulative graph showing the log2 fold-change value of DEGs in dKAT3-ifKOs (mRNA-seq, 1 month after TMX, $n = 3$ per genotype). Upregulated genes are presented in red and downregulated genes in blue (p.adj < 0.05 and |log2FC| ≥ 1). The bottom bar graph compares the area in each set. **b** Volcano plots of RNA-seq analysis. From left to right, we present all genes (left) and the subset of housekeeping genes listed in ref. [76] (right). Gray: genes that are not significantly deregulated; red: upregulated genes; blue: downregulated genes; purple: housekeeping genes. **c** The 10 most enriched categories identified by Gene Ontology (GO) analysis on up- (red) and downregulated (blue) genes with a p.adj < 0.05 and |log2FC| ≥ 1. The upregulated gene set only retrieved three categories. **d** RNA-seq profiles for three representative neuronal-specific genes: *Rbfox3* (NeuN), *Hpca*, and *Camk4*. Scale: 2 kb. **e** Immunohistochemistry against the neuronal protein encoded by *Camk4*, *Rbfox3* (NeuN), and *Hpca*. Dashed line labels the position of the *stratum pyramidale* based on DAPI images ($n = 3$). Scale: 60 μm. **f** Loss of NeuN immunoreactivity in DG neurons infected with a cre recombinase-expressing AAV (GFP+). The experiment was performed 3 times with different sets of mice. The virus was injected unilaterally into the DG of adult *Crebbp*[f/f]::*Ep300*[f/f] mice and the mice were perfused 1 month later (Supplementary Fig. 3e). See Supplementary Fig. 3f for immunostaining against CBP in the same brain slide. Scale: 200 μm. **g** Scheme representing the strategy to eliminate both KAT3 proteins in hippocampal PNCs from E17 dKAT3[f/f] embryos. **h** Representative images showing morphological changes in *Crebbp*[f/f]::*Ep300*[f/f] hippocampal neurons infected with LV-CRE compared with LV-GFP control ($n = 6$ in both groups). **i** RT-PCR demonstrates decreased levels of *Neurod2* (ND2), *Hpca* (hippocalcin), and *Rbfox3* (NeuN) transcripts in dKAT3-KO PNCs. In contrast, several housekeeping genes (*Ppl1*, *Pgk1*, *Rpl23*) are unaffected ($n = 5$–6 in both groups). Data are presented as mean values ± SEM. Two-tailed t-test: ****$p < 0.0001$, ***$p < 0.001$, **$p < 0.01$. Source data for graphs in panel (**i**) are provided as a Source data file.

ChIP-seq) in sorted NeuN+ hippocampal nuclei[22] to discriminate between neuronal-specific, non-neuronal-cell-specific and "pancellular" KAT3 binding (Fig. 4c, d, Supplementary Fig. 10a, and Supplementary Data 3). Intriguingly, pancellular KAT3 peaks are found in the promoter of genes involved in basic cellular functions, such as RNA processing and metabolism, while cell-type-specific peaks (neuronal and non-neuronal) primarily locate at introns and intergenic regions with enhancer features (Supplementary Fig. 10b) and associate with cell-type-specific processes (Fig. 4e, f and Supplementary Data 4). We used binding and expression target analysis (BETA)[23] to integrate ChIP-seq and RNA-seq data (Fig. 4g), and found that the loss of neuronal KAT3 binding (~14,000 peaks) is an excellent predictor of transcriptional downregulation ($p = 3.3 \times 10^{-72}$, Fig. 4h). Up to 74% of the downregulated genes in dKAT3-ifKOs are linked to the loss of KAT3s at proximal regulatory regions. Comparison of ATAC-seq profiles of control and dKAT3-ifKO neuronal nuclei retrieved more than 6000 regions with differential accessibility (Supplementary Data 5). Most of these differentially accessible regions (DARs) displayed reduced accessibility in dKAT3-ifKO neurons (Supplementary Fig. 10c), were located at enhancers (Supplementary Fig. 10d) and coincided with the downregulation of the proximal gene (Supplementary Fig. 10e).

**KAT3s preserve acetylation of neuron-specific enhancers.** In agreement with a recent acetylome analysis[9], immunostaining against specific lysine residues in the histone tails revealed their variable dependence on CBP/p300 (Supplementary Fig. 11a) as well as the particular sensitivity of H3K27ac to the loss of CBP and p300 (Fig. 5a). This histone modification is enriched in active enhancers and its levels correlate with tissue specification[24]. The dramatic reduction in H3K27ac was not accompanied by an increase in the signal or changes in the distribution of H3K27me3 (Supplementary Fig. 11b). H3K9me3, a histone modification associated with heterochromatin, also seemed unaffected (Supplementary Fig. 11c).

We next investigated the genomic distribution of H3K27ac in hippocampal chromatin of dKAT3-ifKOs and control littermates. We retrieved 37,732 H3K27ac-enriched regions (Fig. 5b) that largely overlap with KAT3 peaks (Supplementary Fig. 11d). More than one-third of these H3K27ac peaks were strongly reduced in dKAT3-ifKOs (Supplementary Data 6), particularly those that overlap with neuronal KAT3 binding (~80%). In contrast, less than 5% of pancellular and non-neuronal H3K27ac peaks were affected (Fig. 5b). To explore in greater detail these differences, we classified neuronal KAT3 peaks into promoters and enhancers based on their location and H3K4me1/me3 content (Fig. 5c and

Supplementary Fig. 11e). KAT3 binding, H3K27ac and ATAC-seq signals were all strongly reduced in the enhancers (Fig. 5d) and promoters (Supplementary Fig. 11f) of neuronal genes in dKAT3-ifKOs, although the changes were more abundant and significant for enhancers than promoters (Supplementary Fig. 11g). In contrast, the promoters associated with pancellular-KAT3 peaks were spared (Supplementary Fig. 11f), which suggests that other KATs contribute to maintain H3K27ac at the promoter of pancellular genes. The acetylation of H3K9,14 that primarily decorates promoters only showed a modest reduction in dKAT3-ifKOs (Supplementary Fig. 11h), indicating that other KATs are responsible for maintaining this form of acetylation. We also observed a strong correlation between the loss of H3K27ac and transcript downregulation. Out of 1376 genes downregulated in dKAT3-ifKOs, 78% showed a strong reduction in H3K27ac. Reciprocally, 74% of the genes with reduced acetylation were neuronal genes with reduced expression in dKAT3-ifKOs.

Super-enhancers constitute a special type of regulatory region that plays a critical role in controlling cell identity[25]. Their conspicuous features include the association with highly transcribed genes, broad domains of H3K27 acetylation, and a high density of TF binding sites. To identify putative neural super-enhancers, we fused together enhancers that are closer than 5 kb from each other (Supplementary Data 7). The retrieved regions were associated with highly expressed genes in hippocampal neurons that are downregulated in dKAT3-ifKOs (Fig. 5e). Furthermore, more than 85% of these super-enhancers-associated genes were strongly hypoacetylated in dKAT3-ifKOs (Fig. 5f) and encoded bona fide neuronal regulators related with synaptic transmission and neuronal plasticity (Fig. 5g). Together, these results show that KAT3 proteins control the status of neuron-specific genes by regulating lysine acetylation levels at enhancers and super-enhancers.

**bHLH TFs drive KAT3 binding to neuron-specific genes.** Since KAT3 proteins do not directly bind to DNA, we next asked which proteins are responsible for recruiting CBP and p300 to these neuron-specific regulatory regions. Motif prediction analysis on KAT3 binding peaks revealed remarkable differences between pancellular and neuronal-specific regions. Pancellular KAT3 binding was associated with general TFs such as Sp, Fox, and Ets, all of which have been reported to bind CBP or p300[8] (Fig. 5h). In contrast, neuronal KAT3 peaks presented a very prominent enrichment ($E$-value = $1.0 \times 10^{-1223}$) for the DNA binding motif of basic helix–loop–helix (bHLH) TFs (Fig. 5h). Regions losing

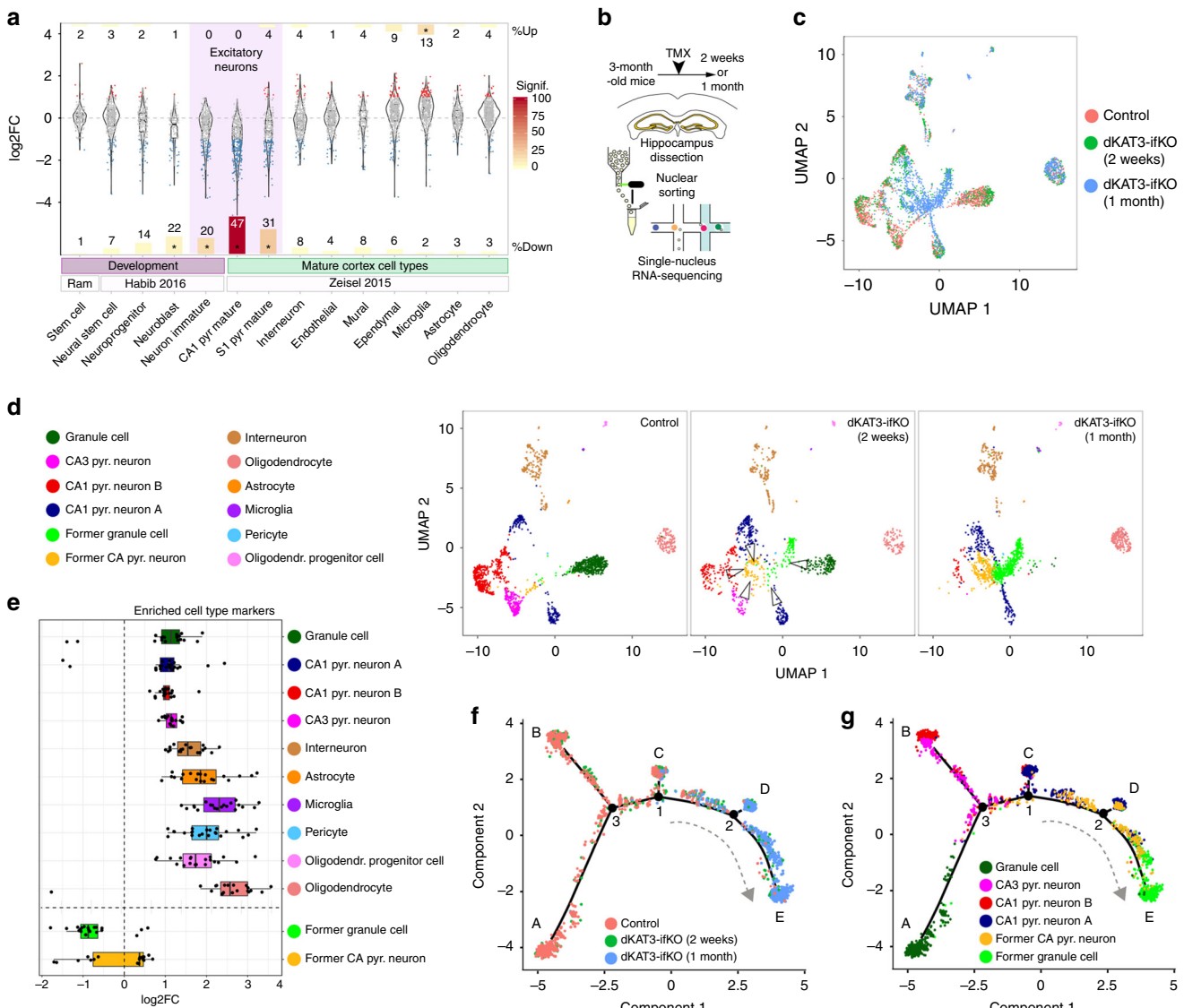

**Fig. 3 Cells lacking KAT3 proteins do not die or dedifferentiate, but acquire a novel, molecularly undefined fate. a** Violin plots show the change in expression of gene sets associated with stemness[19] and different neuronal differentiation stages[20] during development (purple bar) or with different cell types in the adult mouse cortex[18] (green bar). Each dot represents a single gene. Number of genes tested per cell type: stem cell = 195, neural stem cell = 365, neuroprogenitor = 200, neuroblast = 109, immature neuron = 353, CA1_pyr mature = 371, S1_pyr mature = 251, interneuron = 337, endothelial = 337, mural = 141, ependymal = 420, microglia = 387, astrocyte = 223, oligodendrocyte = 418. Bar sizes are proportional to the percentage of up- or downregulated genes in each gene set as indicated by the number in each bar. Bar colors indicate the significance of the enrichment (−log10) in a hypergeometric test for the number of regulated genes. Enrichments with p-values < 5e-10 are labeled with an asterisk (p-val_Immature_neuron = 4.82e-17; p-val_CA1_Pyramidal = 2.59e-101; p-val_S1_Pyramidal = 4.69e-25; p-val_Microglia = 3.49e-15). **b** Scheme of the single-nucleus RNA-seq experiment. **c** UMAP plot of integrated analysis of snRNA-seq datasets from the dorsal hippocampus of dKAT3-fKOs and control littermates. **d** UMAP plots showing identified populations in the hippocampus of control littermates (left) and dKAT3-ifKO mice 2 (center) and 4 (right) weeks after TMX treatment. Nuclei are colored by their classification label as indicated. **e** Boxplots showing the expression of the top 20 markers for the 10 cell types detected in the hippocampus of control mice and the two new clusters detected in dKAT3-ifKO mice 1 month after TMX (Supplementary Data 2). Whiskers lengths are 1.5 the interquartile range from the box. Abbreviations: Pyr, pyramidal; Oligodendr, oligodendrocyte. **f**, **g** Single-nucleus trajectory analysis of hippocampal excitatory neurons reveals cell state-transitions toward a non-functional interstate deadlock. Nuclei are colored by experimental condition (**f**) or cluster subpopulation (**g**) as indicated in the legends. Loss of dKAT3s caused the progressive relocation of the cells from the root in the branches A, B, and C (expressing markers for different types of excitatory neurons) towards the outcome in branches D and E that are populated by the type of cells described in (**e**) that do not express distinctive markers. Source data for graphs in panels (**a**) and (**e**) are provided as a Source data file.

occupancy in dKAT3-ifKOs were also highly enriched for bHLH-bound motifs (Fig. 5i and Supplementary Fig. 12a).

Among these bHLH TFs, there are important pro-neural TFs involved in neuronal development[26,27] (e.g., *Ascl1*, *Neurod1-6*, *Neurogenin1-3*, and *Atoh1*). Our differential expression analysis retrieved 20 bHLH TF-encoding genes that were downregulated

in the hippocampus of dKAT3-ifKOs (Supplementary Fig. 12b and Supplementary Data 1). In fact, *NeuroD2* and *NeuroD6*, which encode TFs with putative terminal selector activity[4,28], were among the most downregulated genes in the hippocampi of dKAT3-ifKOs (Fig. 5j). To directly assess the occupancy of these sites before and after the ablation of KAT3s, we analyzed their

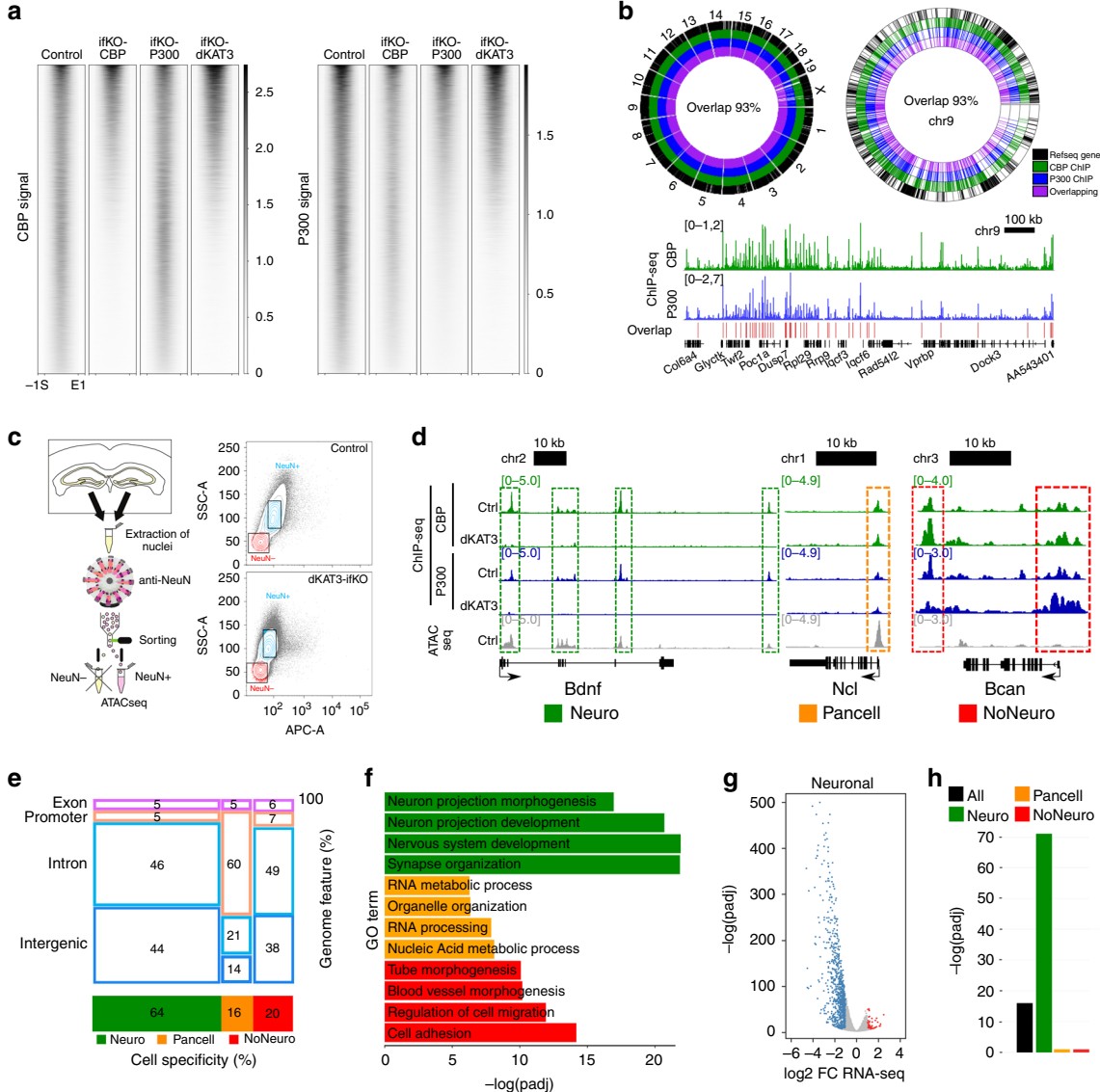

**Fig. 4 CBP and p300 bind to the same genomic sites. a** Heat maps showing the control CBP/P300 KAT3 ChIP-seq peaks and the signal in the corresponding locations of CBP-, p300-, and KAT3-ifKO. Intensity ranges from strong (black) to weak (white). S: peak start, E: peak end, ±1 kb. **b** Circos plots of the entire genome (left) and chromosome 9 (right). Below: a snapshot of a gene-rich region in chromosome 9. Colors indicate CBP and p300 binding in hippocampal chromatin and their overlap. Refseq genes in black. **c** Left: scheme of the FANS/ATAC-seq experiment. Right: flow cytometry sorting plots for control and dKAT3-fKO samples. Boxes indicate the gates used for sorting NeuN+ nuclei (blue). APC-A is the signal of anti-NeuN staining. **d** Snapshot illustrating the classification of KAT3 peaks in neuronal (green), non-neuronal (red), and pancellular (orange). Representative peaks classified as neuronal, non-neuronal, and pancellular are marked with a green, red, and orange dashed rectangle, respectively. **e** Classification of KAT3 peaks according to cell specificity (neuronal, pancellular, and non-neuronal) and genomic features (promoter, exon, intron, intergenic). The numbers within each sector represent percentages. **f** GO enrichment analysis performed on the gene sets associated with neuronal (green), non-neuronal (red), and pancellular (orange) KAT3 peaks. **g** Volcano plots of RNA-seq analysis for the subset of genes classified as neuronal. Red: upregulated genes; blue: downregulated genes. **h** BETA analysis of the association between all, neuronal, non-neuronal, and pancellular KAT3 peaks, and transcriptome changes. Neuronal peaks show the strongest association with gene downregulation. *p*-Values in Kolmogorov–Smirnov test using BETA-cistrome software: *p*-val$_{All}$ = 6.24e−17; *p*-val$_{Neuro}$ = 3.34e−72; *p*-val$_{NoNeuro}$ = 0.3; *p*-val$_{Pancell}$ = 0.3.

digital footprints in DARs and detected robust differences (Fig. 5k and Supplementary Fig. 12c), confirming that some changes in chromatin occupancy reflect the reduced binding of members of this family. This finding is consistent with the large overlap between bHLH TF footprints detected in the ATAC-seq profiles and Neurod2 ChIP-seq data[29] (Supplementary Fig. 12d). Altogether these experiments indicate that CBP and p300 interact with bHLH proneural TFs in neuronal-specific genomic locations to maintain neuron-specific transcription (Fig. 5l and Supplementary Fig. 12e show some representative examples).

**Other cell type fates also require KAT3 proteins.** Previous studies have shown that KAT3 proteins play a critical role in the differentiation of several cell types and the establishment of cell-type-specific transcription[30–33]. To investigate if cell fate maintenance in other cell types also requires the KAT3 activity, we next examined astrocytes derived from dKAT3-floxed mice (Fig. 6a). We prepared astrocyte cultures from the hippocampi of *Crebbp$^{f/f}$::Ep300$^{f/f}$* embryos and infected them with a Cre-recombinase-expressing lentivirus. Similar to our results in neurons, primary cultures of cortical astrocytes missing both KAT3

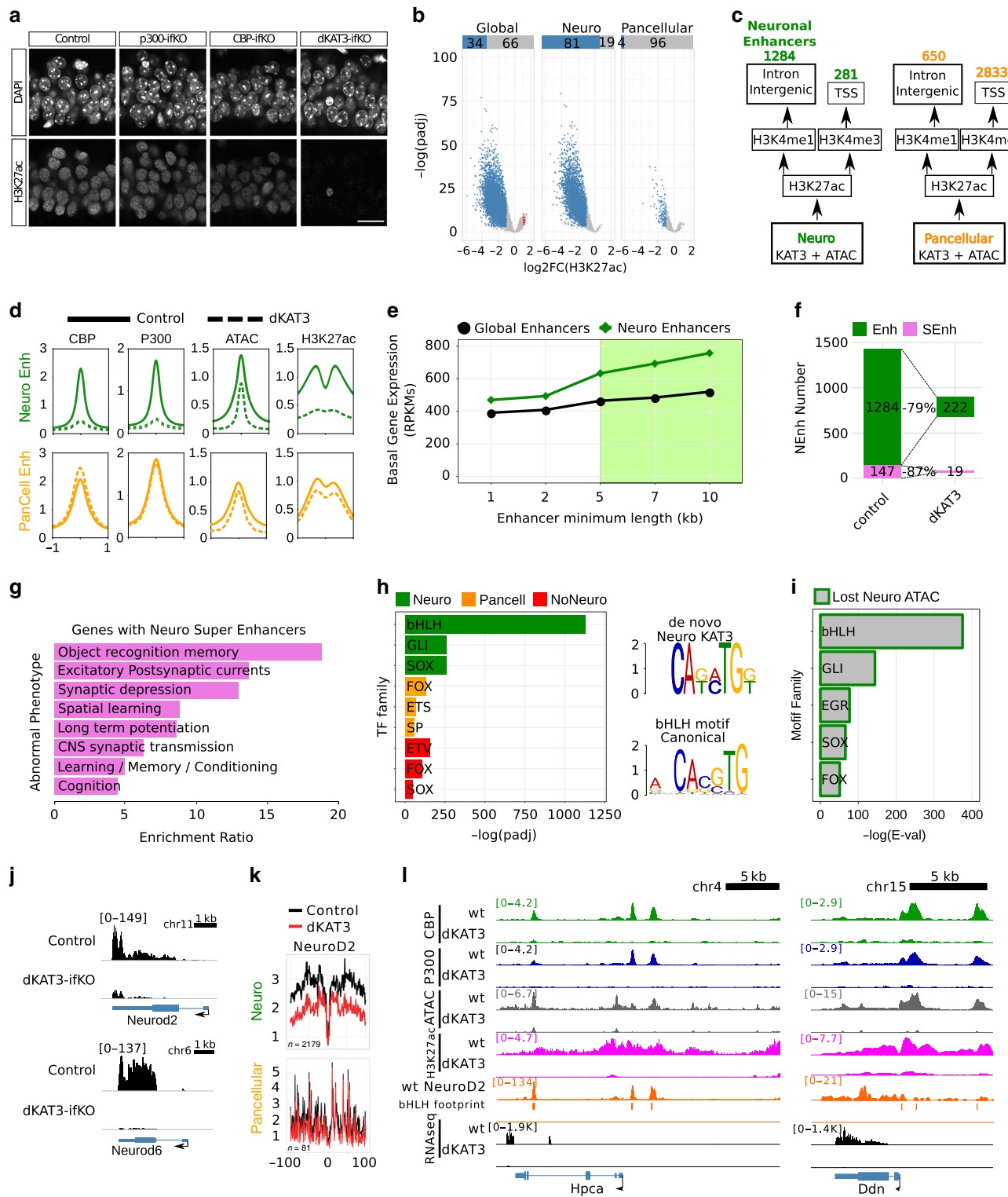

proteins lose both the expression of glial genes such as *Gfap* and their characteristic morphology (Fig. 6b–d), indicating that KAT3 proteins are also responsible for identity maintenance in other cell types.

To examine this possibility in cells from other lineages, we investigated the profiles of p300 binding in different tissues available at ENCODE (there are no tissue-specific profiles for CBP). We found that, similarly to our observations in brain tissue, p300 peaks were associated with genesets highly enriched in tissue-specific functions (Supplementary Fig. 13a). Furthermore, in each tissue, p300-bound regulatory regions were enriched for distinct binding motifs (Supplementary Fig. 13b) and tissue-specific p300 binding was observed into and upstream of loci presenting tissue-specific transcription (Supplementary Fig. 13c). Thus, we detected a very high enrichment for GATA TFs in liver. TFs belonging to this family are essential during liver development and are still expressed in mature tissue[34], indicating that they may work as terminal selectors for hepatocytes.

**Fig. 5 H3K27ac is strongly decreased in neuro-specific locations and correlates with gene downregulation. a** Immunostaining against H3K27ac in the CA1 subfield of single and dKAT3 ifKOs (IF signal control: 21.62 ± 2.24, dKAT3-ifKOs: 4.94 ± 1.03, $p < 0.0001$, unpaired $t$-test); the single ifKOs show non-significant difference compared to controls. The experiment was repeated 3 times. Scale: 10 μm. **b** Genome-wide analysis of H3K27ac changes in dKAT3-ifKOs. Volcano plots show the fold change and significance values for all peaks (Global) and for the subsets of neuron-specific (Neuro) and pancellular peaks. **c** Classification of regulatory regions categorized by genomic features. **d** Metaplots of ATAC-seq, KAT3, and H3K27ac ChIP-seq signals in neuronal and pancellular enhancers in controls and dKAT3-ifKOs. Plots are centered in the peak center and expanded ±1 kb. **e** Correlation between length of the enhancer and expression level of the proximal gene. **f** Number of downregulated genes that contain enhancers or super-enhancers in control mice and percentage that lose acetylation in dKAT3-ifKOs. **g** Barplot showing the top 10 enriched categories from the Phenotype analysis for genes harboring neuronal enhancers. **h** TFBS analysis of neuronal (green), non-neuronal (red), and pancellular (orange) KAT3 peaks. Each motif family name is a user-curated approximation to the results provided by the MEME-suite algorithm. The most enriched de novo-identified binding motif in neuronal KAT3 (upper right motif) is very similar to the NeuroD2 motif (bottom right). Log $E$-values obtained with the expectation maximization algorithm (MEME-ChIP suite): Neuro: bHLH = 1130, GLI = 258, SOX = 257; Pancell: FOX = 130, ETS = 67, SP = 57; NoNeuro: ETV = 157, FOX = 105, SOX = 49. **i** TFBS analysis of neuronal regions with a reduced ATAC-seq signal in dKAT3-ifKOs. **j** RNA-seq profiles for *Neurod2* and *NeuroD6*. Scale: 1 kb. **k** Digital footprinting of NeuroD2 at neuronal and pancellular KAT3-bound regions. Values correspond to normalized Tn5 insertions. **l** Representative snapshots of KAT3, ATAC, and H3K27ac depletion with bHLH footprint and NeuroD2 overlaps, at two genes that are strongly downregulated in dKAT3-ifKOs. Source data for graphs in panels (**e**) and (**h**) are provided as a Source data file.

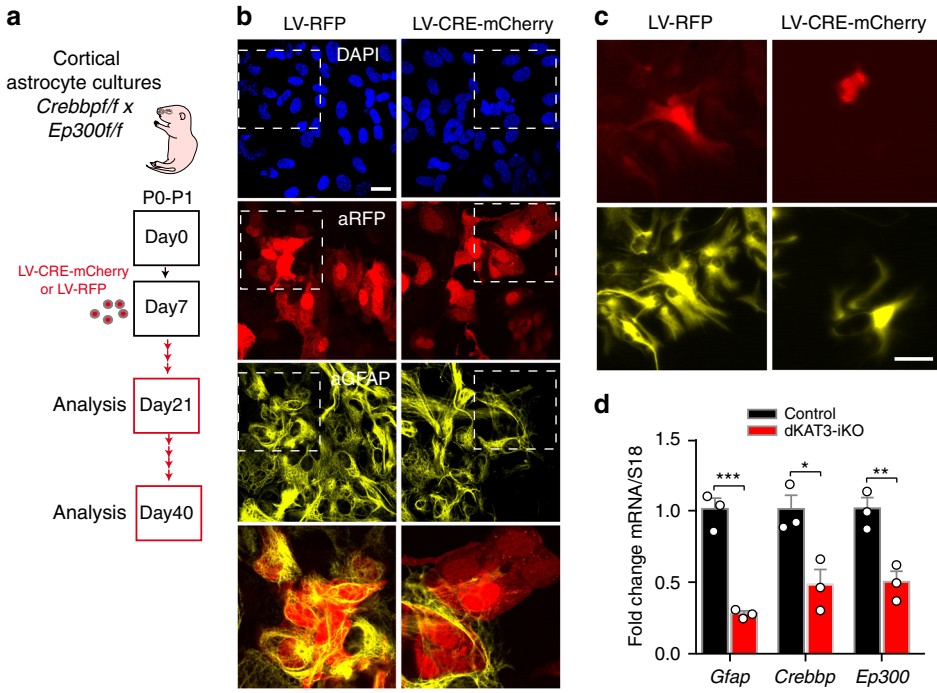

**Fig. 6 CBP and p300 are needed to maintain the fate of other cellular types. a** Generation of dKAT3-KO astrocytes. Cortical astrocytes from *Crebbp*f/f:: *Ep300*f/f pups were infected with a cre recombinase-expressing LV. **b** Cultured dKAT3f/f astrocytes show a downregulation of astrocyte marker GFAP 2 weeks after infection with a cre recombinase-expressing LV ($n = 3$ for both groups). Scale bar: 20 μm. **c** The loss of astrocyte morphology is more evident 4 weeks after infection (n = 3 for both groups). Scale bar: 50 μm. **d** RT-PCR quantification of the *Gfap* (GFAP), *Crebbp* (CBP), and *Ep300* (p300) transcript levels in cultured dKAT3f/f astrocytes ($n = 3$). Data are presented as mean values ± SEM. Two-tailed $t$-test: ****$p < 0.0001$, ***$p < 0.001$, **$p < 0.01$; *$p < 0.05$. Source data are provided as a Source data file.

**Scaffold and KAT activities are both required**. To explore the specific contribution of the scaffolding and KAT activities of KAT3 proteins to the loss of cell-type-specific transcription, we turned to PNCs. Transfection of a heterologous full-length CBP in neurons that have lost the endogenous expression of both CBP and p300 (Fig. 7a, b) prevented the downregulation of the neuronal markers NeuroD2 and hippocalcin (Fig. 7c, d). We next examined whether the expression of the N-terminus or the C-terminus (bearing the KAT domain) halves of CBP, as well as the full-length reconstituted protein (Fig. 7e, f) could also rescue the transcriptional impairment. To specifically assess the contribution of KAT activity, we also transfected a variant of the C-terminus half of CBP (referred to as KATmut) bearing the R1378P mutation linked to RSTS[35]. We found that only the

reconstruction of full-length CBP with an intact KAT domain prevented the downregulation of the neuronal marker NeuroD2 (Fig. 7g, h). These results indicate that both activities of CBP, as molecular scaffold and KAT, are necessary to preserve the active status of target loci.

We also investigated whether heterologous NeuroD2 expression exerted the same protection. This bHLH TF is highly expressed in mature excitatory neurons, regulates its own expression, and holds the features of a neuronal terminal selector[4]. Furthermore, it is strongly downregulated in dKAT3-ifKOs neurons (Fig. 5j and Supplementary Fig. 12b), and its occupancy profile in cortical chromatin[29] shows a large overlap with DARs in dKAT3-ifKOs (Supplementary Fig. 14a, b). Conversely to CBP, NeuroD2 alone was not sufficient to re-

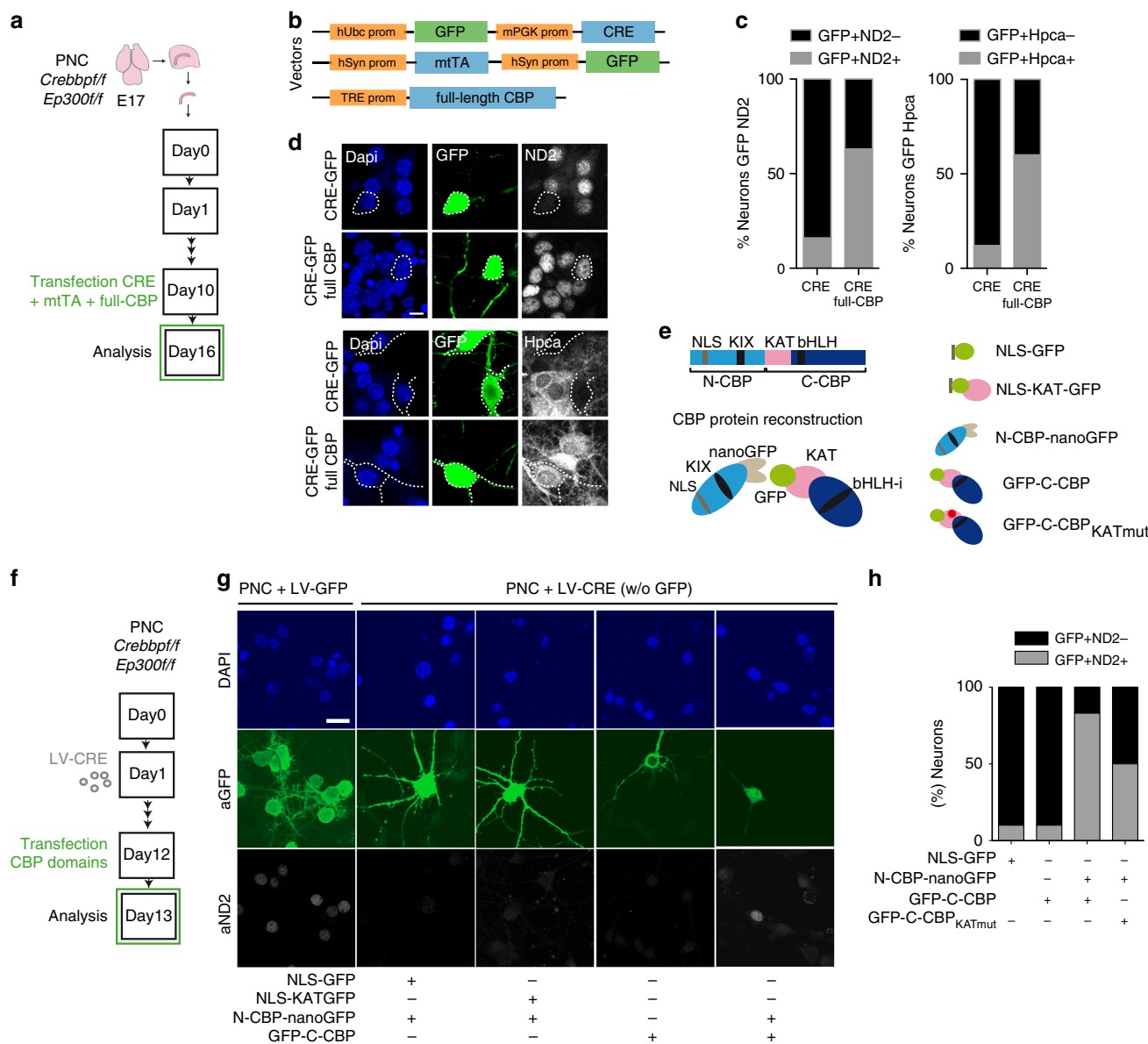

**Fig. 7 Full-length CBP is required to restore neuronal-specific transcription. a** Hippocampal PNCs from E17 *Crebbp*^f/f^::*Ep300*^f/f^ embryos were co-transfected with constructs that drive the expression of the Cre recombinase and full-length CBP. **b** Plasmid combination to express recombinant CBP simultaneously to endogenous CBP and p300 ablation. **c** Representative images of NeuroD2 and hippocalcin staining in PNC transfected with the constructs shown in (**b**). Note the reduced expression in the GFP+ cells in the absence of heterologous full-length CBP (experiments in two independent PNCs). Scale: 10 μm. **d** Quantification of the percentage of NeuroD2-positive or -negative cells and hippocalcin-positive or -negative cells among all GFP-positive cells (*n* = 60 neurons per condition). **e** Scheme of the CBP fragments used for rescuing NeuroD2 expression (see Methods for additional details). NLS: nuclear localization domain; KAT: acetyltransferase domain; KIX: kinase-inducible domain interacting domain; bHLH-i: region of interaction with bHLH transcription factors. **f** Hippocampal PNCs from E17 *Crebbp*^f/f^::*Ep300*^f/f^ embryos were transfected with constructs that drive the expression of the Cre recombinase and the different CBP fragments and the KAT domain shown in panel (**e**). **g** Representative images of NeuroD2 staining in PNCs infected with LV-CREw/oGFP and transfected with the different domains of CBP shown in panel (**e**). PNCs infected with LV-GFP (i.e. with wild type phenotype) were used as a control for baseline NeuroD2 level (2 independent PNCs). Scale: 20 μm. **h** Quantification of the percentage of NeuroD2-positive or -negative cells among all GFP-positive neurons (*n* = 30–80 neurons per condition). Source data for graphs in panels (**d**) and (**h**) are provided as a Source data file.

establish neuronal-specific transcription in dKAT3-KO PNCs (Supplementary Fig. 14c).

**Locus-specific epi-editing rescue transcriptional impairment.** Next, we examined whether increasing lysine acetylation is sufficient to rescue the transcriptional deficit. To this end, we took advantage of recently developed tools for epi-editing based on the expression of an inactive Cas9 enzyme (dCas9) fused to the KAT domain of p300 (dCas9-KAT)[36]. This system enables targeting

specific genomic regions for p300-dependent acetylation using adequate guide RNAs (gRNA). We selected *Neurod2* as the target gene because it is both severely downregulated and H3K27-hypoacetylated in dKAT3-ifKOs (Fig. 8a). The infection of PNCs with lentiviruses that express dCas9-KAT and a gRNA that recruits this chimeric protein to the most proximal KAT3 peak of *Neurod2* (Fig. 8a, b) prevented the downregulation of NeuroD2 observed in dKAT3-KO neurons (Fig. 8c, d). Moreover, co-transfection of dCas9-KAT and the *Neurod2* gRNA in dKAT3-

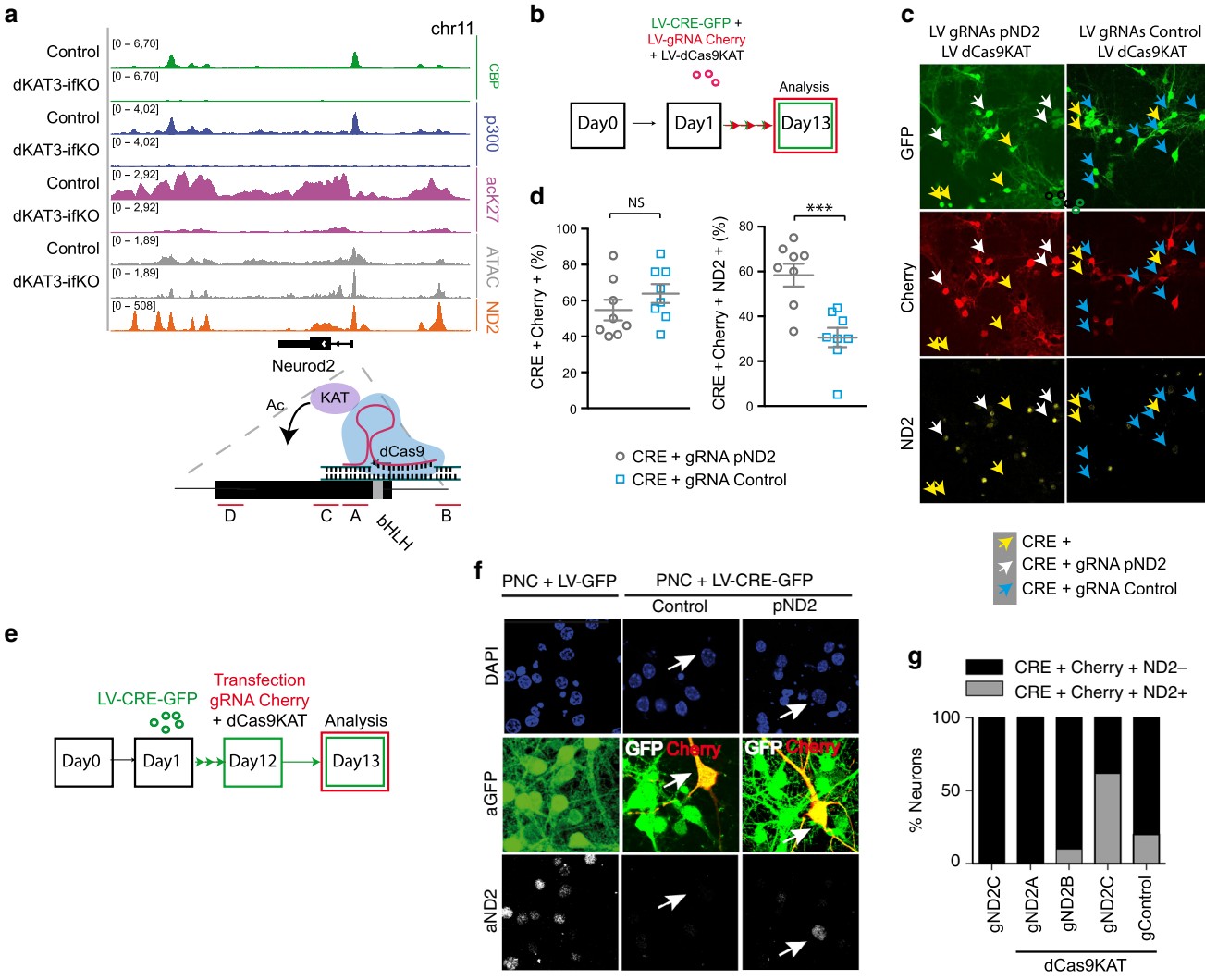

**Fig. 8 Locus-specific acetylation restores NeuroD2 transcription. a** Snap view of the Neurod2 locus. The profiles for CBP and p300 binding, H3K27 acetylation and ATAC-seq signal are shown. The bottom orange track corresponds to the NeuroD2 ChIP-seq data generated in ref. [29]. A scheme presenting the strategy used to drive the KAT activity of p300 to the *Neurod2* promoter using a fusion protein with dCas9 is also shown. The positions targeted by the gRNAs (red lines, gND2 A-D) are indicated. Note that the target regions are in the proximity of bHLH sites. **b** Scheme of the co-infection of LVs expressing cre recombinase, dCas9-KAT, and the Neurod2 gRNAs (A-D mix). **c** Representative image of NeuroD2 protein levels in the cells co-infected with LV-CRE-GFP and the lentiviruses LV-gRNA-mCherry specific for *Neurod2* (white arrows). As a specificity control, we conducted the same experiment using a gRNA targeted to *Hpca* (blue arrows) ($n = 4$ wells per condition in 2 PNCs). As a comparison, cells infected with LV-CRE-GFP alone (yellow arrows) show strongly diminished NeuroD2 levels as well. Scale bar: 50 μm. **d** Quantification of different cell subpopulations observed in the experiment shown in panel (**c**) ($n = 4$ wells per condition in 2 PNCs). Two-tailed *t*-test: ****$p < 0.0001$, ***$p < 0.001$, **$p < 0.01$; *$p < 0.05$. Data are presented as mean values ± SEM. **e** Scheme of rescue experiment with plasmids carrying dCas9-KAT and the Neurod2 gRNA in dKAT3-KO cells that had already lost Neurod2 expression. **f** Representative image of NeuroD2 expression in control (PNC + LV-GFP) and dKAT3-depleted (PNC + LV-CRE-GFP) neurons after transfection with dCas9-KAT and gND2-C targeting the NeuroD2 promoter. Arrows indicate LV-CRE-GFP infected cells transfected with the gRNA-carrying vector. A gRNA targeting the hippocalcin promoter was used as a specificity control. Scale: 20 μm. **g** Quantification of the percentage of transfected cells showing normal NeuroD2 expression after transfection with gND2-C alone and dCas9-KAT co-transfected with gND2-C, gND2-A, gND2-B, or gRNA control independently (experiments in three independent PNCs; $n = 30–40$ neurons per condition). Source data for graphs in panels (**d**) and (**g**) are provided as a Source data file.

KO cells that had already lost Neurod2 expression also caused a significant recovery of the expression of this gene (Fig. 8e–g). These experiments indicate that histone/lysine deacetylation is the main cause for the downregulation of neuronal genes after KAT3 loss. We noted that transcriptional rescue was observed in *Neurod2*, which is a gene that had not lost chromatin accessibility at their promoter, but not in *Hpca*, which showed a severe loss of accessibility (compare ATAC-seq profiles in Figs. 5l and 8a). This suggests that the efficacy of the rescue also depends on other features of the locus.

## Discussion

The differentiation of neuronal lineages during brain development requires the participation of TFs and chromatin-modifying enzymes such as CBP and p300[5,10,37–41]. These proteins are often referred to as gatekeepers of cell fate. Here, we used an inducible knockout system to demonstrate that KAT3 proteins are jointly required in excitatory neurons to preserve their postmitotic neuronal identity, thereby acting as fate-keepers. This evidence indicates that the epigenetic landscape of neuronal genes is not self-sufficient and requires the active and continuous presence of

KAT3 enzymes, which opens up the possibility of using controlled KAT3 inhibition for dedifferentiation and reprogramming in cell therapy strategies. Supporting this view, a recent study showed that treatment with competitive inhibitors targeting the bromodomains of CBP and p300 enhanced the reprogramming of human fibroblasts into iPSCs, while KAT inhibition prevented iPSC formation[42] (which is consistent with our observation that dKAT3-KO cells fail to express stem cell markers).

Strikingly, the loss of neuronal identity did not lead to apoptosis or other forms of neuronal death even weeks after KAT3 elimination both in culture and in vivo. This may be the result of the inability of dKAT3-KO cells to trigger the programmed cell death program. As shown in Supplementary Fig. 4b, numerous genes involved in neuronal apoptosis and death are strongly downregulated in dKAT3-KO, including important initiators of neuronal apoptosis such as *Hrk*[43,44]. This gene is highly expressed in neurons and as many other neuronal genes present a dramatic downregulation that is accompanied by the loss of CBP/p300 binding and the H3K27 deacetylation of the locus (Supplementary Fig. 11e). Together these results suggest that KAT3 proteins are essential to safeguard cell identity but also to activate any alternative cell fate, including stemness and programmed cell death.

Thanks to their ability to interact with cell-type-specific TFs, KAT3 proteins are recruited to specific genomic locations where they act as an "acetyl-spray" targeting accessible lysine residues on proximal histones and non-histone proteins[9] to support active transcription[10]. Interestingly, a single *Crebbp* or *Ep300* functional allele is sufficient to sustain neuronal identity, which underscores the robustness of the epigenetic mechanisms involved in cell type maintenance. Robustness and redundancy are even greater in ubiquitous and highly expressed genes for which we detected neither downregulation nor severe hypoacetylation in dKAT3-ifKOs, likely because other KATs maintain the acetylation at these loci. Consistent with this hypothesis, our RNA-seq screen shows that KATs such as Gcn5/KAT2A, Tip60/KAT5, and MOF/KAT8, which are expressed in neurons and localize to promoters[45,46], are not downregulated in dKAT3-ifKO neurons (Supplementary Fig. 11e shows the profile of *Kat5*; a similar pattern is observed in *Kat2A* and *Kat8*). The recent discovery that KAT5, contrary to KAT3 proteins, is required for neuronal viability[47] further supports an essential role for that KAT in cell homeostasis.

Our experiments show that the recruitment of KAT3 proteins to cell-type-specific enhancers in excitatory hippocampal neurons is likely mediated by bHLH TFs belonging to the NeuroD family. Various bHLH TFs are differentially expressed throughout neuronal proliferation and specification. For instance, Hes1 and Hes5 promote NSC/NPC renewal and inhibit specification, but are replaced by Ascl1 or Neurog2 to trigger the differentiation to cortical neurons[48]. These two TFs are followed or accompanied by other factors like NeuroD1 and other family members, such as NeuroD2 or NeuroD6, whose expression is maintained throughout the lifetime of the neuron[49]. Although we cannot be certain which specific bHLH TF or set of TFs actually recruit CBP and p300 in adult neurons, the two KATs are known to interact with multiple bHLH TFs including MyoD[50], Ascl1[51], Neurogenins[37,52], Twist[53], and NeuroD2[54]. Since bHLH proteins form dimers[48], and the binding to a dimerized TF has recently been shown as essential for the control of KAT3 activity[55], these TFs represent suitable candidates to regulate KAT3 function and drive chromatin acetylation at different stages of neurodevelopment.

The loss of neuronal identity observed in the brain of dKAT-ifKOs is unlikely to be a mere consequence of the downregulation of pro-neural TFs because this phenotype was not observed after elimination of *Neurod1*, *Neurod2*, *Ascl1*, or other bHLH genes in neurons[49,56], nor was this phenotype rescued by NeuroD2 overexpression. It is also unlikely to be the direct consequence of a general loss of the enhancer's building factors because our differential gene expression analysis indicates that, besides the two KAT3 proteins, the components of the enhanceosome[57] are not downregulated in dKAT3-ifKOs. Instead, our experiments show that identity loss is strongly associated with chromatin hypoacetylation and that it can be avoided by targeted KAT activity. These results suggest that the main role of terminal selectors might be to recruit the KAT3 enzymes for maintaining the acetylated status of cell-type-specific enhancers in differentiated cells. These findings have broad clinical implications because impaired CBP/p300 function, histone hypoacetylation, and the loss or attenuation of epigenetic profiles underlying cell fate are features of several neurological disorders, including Alzheimer's and Huntington's diseases and aging-related senescence[58,59].

Importantly, the role of KAT3 proteins as fate-keepers is not restricted to neurons. Our experiments in astrocytes, analysis of KAT3 peaks in different tissues and previous observations in other cell types demonstrating that the combined loss of CBP and p300 causes impaired cell-type-specific transcription[30–33], all support this view. Further supporting this model, a very recent study in mice with skeletal muscle-specific and inducible combined ablation of p300 and CBP revealed that the two proteins are also jointly required for the control and maintenance of contractile function and transcriptional homeostasis in skeletal muscle[60]. What may differ between cell types is the specific set of TFs that recruit the KAT3 proteins to cell-type-specific loci. Future studies should identify their cell-type-specific partners and targets governing tissue specification.

The role of KAT3 proteins in preserving cell-type-specific gene statuses can be particularly relevant for neurons given their tremendous diversity and long lifespan. The critical importance of KAT3s for brain function may explain the duplication of the ancestral KAT3 gene in the first vertebrates coinciding with the emergence of a neural crest and cephalization[10,61]. Although the two KAT3 proteins have evolved some individual functions in postmitotic cells (e.g., forebrain restricted CBP knockouts show phenotypes related to cognitive dysfunction[58] and p300 seems to play a prominent role in muscle biology[62]), our study demonstrates that they still share a joint and more essential role in preserving epigenetic identity.

## Methods

**Animals and treatments**. Animals were housed according to the Spanish and European regulations and the experiments were approved by the Animal Welfare Committee at the Instituto de Neurociencias and the CSIC Ethical Committee. Mice were caged in ventilated cages in a pathogen-free facility with 12 h light/dark cycle, food and water available ad libitum, and controlled temperature (23 °C) and humidity (40–60%). Both male and female mice were used in the experiments. CaMKIIα-creERT2[17], *Ep300*[f/f][63], and *Crebbp*[f/f][64] mice were crossed to obtain p300-ifKO, CBP-ifKO, and dKAT3-ifKO mice. These CaMKIIα-creERT2 carrying mouse lines were maintained in standard housing for 3–4 months. Then, they were treated with tamoxifen (TMX)[65] to trigger the elimination of either p300, CBP, or both proteins in forebrain excitatory neurons. CaMKIIα-creERT2[−] littermates were considered as controls. dKAT3-ifKO animals in poor state or showing signs of pain were sacrificed. To improve their survival, we provided dKAT3-ifKOs and their control littermates with high-protein food pellets (Teklad Global Diets® 2919, Envigo) on the bedding.

**Lentiviral production and plasmid constructs**. Lentiviral particles (LV) were produced according to the methods established in our laboratory[66]. The following lentiviral constructs were acquired from Addgene: LV-CRE-GFP (Addgene #20781), LV-CRE-mCherry (Addgene #27546), and LV-RFP (Addgene #17619) were used to label neurons and astrocytes with fluorescent reporter proteins; dCas9-KAT (Addgene #61357), LV-dCas9-KAT (Addgene #83889), and LV-U6gRNA-Cherry (Addgene #85708) were used in epi-editing experiments; and LV-phND2-N174 (Addgene #31822) was used to overexpress human NeuroD2. All the

designed gRNAs were cloned in LV-U6gRNA-Cherry using the oligonucleotides pairs indicated in Supplementary Table 2. The locations of the gRNAs were selected based on the enrichment for CBP, p300, and H3K27ac in the chromatin of dKAT3-iKOs. To produce LV-CREw/oGFP we digested LV-CRE-GFP with XbaI and BsrGI to remove GFP. As a control for Cre recombinase transduction, we infected the neurons with a synapsin promoter-bearing lentiviral vector derived from LenLox 3.7, LV-syn-syn-GFP[67]. The constructs encoding CBP fragments were subcloned by PCR from full-length CBP: N-terminus-CBP-vhhGFP4 spans amino acids (aa) 1–1098; GFP-C-terminus-CBP spans aa 1099–2441; NLS-KATGFP includes a sequence encoding a nuclear localization signal (KKKRKVD) fused to aa 1088–1758 of CBP and to full-length GFP. pDsRed-Express2-C1 (Clontech) was used in the analysis of neuronal morphology in hippocampal cultures.

**Primary cultures and lentiviral infection.** Primary hippocampal and cortical cultures were prepared from $Crebbp^{f/f}::Ep300^{f/f}$ and $Crebbp^{f/f}::Ep300^{f/f}::CAG/loxP/STOP/loxP/tdTomato$ embryos. Their hippocampi were dissected, pooled together, and dissociated for neuronal extraction[66]. Cells were plated in 24-well plates at $0.11 \times 10^6$ neurons/well. After 24 h, the primary cultures were infected with the indicated viruses (day in vitro 1, DIV1). In the experiments preventing CBP loss, neuronal cultures were co-infected at DIV1 with the indicated viruses and the cultures were processed after 11 days. In rescue experiments, neurons were first infected with the Cre recombinase-expressing LV at DIV1, the same neurons were transfected with the indicated vectors at DIV12 and the cultures were fixed at DIV13. To produce the astrocyte primary cultures, astrocytes were isolated from cortices of P1-P3 $Crebbp^{f/f}::Ep300^{f/f}$ pups. The cortices were dissected, cut into pieces, and washed twice with HBBS 1× (Lonza BE10547F). The tissue was disrupted with a Pasteur pipette with rounded edges in 2 ml of complete medium (DMED from Gibco, 21969-035 plus 10% fetal bovine serum and 1% penicillin/streptomycin). The homogenate was filtered with a 70 µm cell strainer (Falcon #352350), centrifuged, and resuspended in complete medium. The astrocytes from two pups were pooled together and plated in 24-well plates. Astrocytes cultures were infected at DIV10 with LV-CRE-mCherry or LV-RFP as a control, and fixed with PFA 4% at DIV17 or DIV30. In all immunohistochemistry experiments, cells were cultured on glass coverslips coated with poly-lysine.

**RNA isolation, RT-qPCR, and transfection.** Total RNA from hippocampal or cortical cultures was extracted with TRI reagent (Sigma-Aldrich) and reverse transcribed to cDNA using the RevertAid First-Strand cDNA Synthesis kit (Fermentas). RT-qPCR was performed in an Applied Biosystems 7300 Real-Time PCR unit or a QuantStudio 3 unit using Eva Green RT-PCR reagent mix and the primer pairs indicated in Supplementary Table 2. The transfections were done using Lipofectamine 2000 (Invitrogen) at DIV12 and the cultures were fixed at DIV13 for morphology analysis and rescue experiments.

**ChIP-assay.** H3K27ac chromatin immunoprecipitation was performed as described[68]. The CBP and p300 ChIP experiments required the following adjustments: minced hippocampal tissue was fixed in 1% PFA for 30 min at 37 °C (as suggested in ref. [69]), which allowed for crosslinking of the KAT3 cofactors to the DNA-binding proteins and to the DNA itself. Because KAT3 proteins are almost completely degraded after a few rounds of sonication in 1% sodium dodecyl sulfate (SDS), the sonication buffer was changed to one containing 0.1% SDS, 1% IGEPAL (Sigma-Aldrich), and 0.5% sodium deoxycholate. Ten additional sonication cycles of 30″ on/30″ off were added to fragment the highly fixed DNA. ChIP-qPCR assays were performed in a QuantStudio 3 unit using the primer pairs indicated in Supplementary Table 2.

**Antibodies.** The following primary antibodies have been used in this study: anti-CBP, Santa Cruz sc-583 (IHC: 1:500; ChIP: 10 µg); anti-CBP, Santa Cruz sc-369 (ICC: 1:100); anti-CBP, Santa Cruz sc-7300 (IHC: 1:100; ICC: 1:100); anti-p300, Santa Cruz sc-585 (IHC: 1:100; ICC: 1:100; ChIP: 10 µg); anti-NeuroD2, Abcam ab109406 (ICC: 1:100); anti-H2Aac[70], (IHC: 1:100); anti-H2Bac[70], (IHC: 1:1000; ICC: 1:1000); anti-H3K9,14ac[70], (IHC: 1:400); anti-H3K27ac, Abcam ab4729 (IHC: 1:1000; ICC: 1:1000; ChIP: 5 µg); anti-H3K27me3 (IHC: 1:100); anti-H3K9me3 ab8898 Abcam (IHC: 1:100); anti-H4ac[70], (IHC: 1:100); anti-NeuN, MAB377 Millipore (IHC: 1:500; FANS: 1:500); anti-Hpca, Abcam ab24560 (IHC: 1:500; ICC: 1:500); anti-CaMKIV, BD Transduction Laboratories C28420 (IHC: 1:500); anti-Cleaved-Cas3, Cell Signalling #9661 (IHC: 1:200); anti-Fos, Synaptic Systems #226004 (IHC: 1:500); anti-mCherry/dsRed, Clontech 632496 (ICC: 1:1000); anti-GFP, Aves Labs GFP-1020 (IHC: 1:500; ICC: 1:1000); anti-GFAP, Sigma G9269 (IHC: 1:200; ICC: 1:100); anti-H2A.Xγ, Abcam ab2893 (IHC: 1:200). Biotinylated anti-mouse (Sigma B0529, 1:500) and anti-rabbit (Sigma B8895, 1:3000) antibodies were used in the DAB staining (Sigma Cat. 11718096001). Fluorophore-coupled secondary antibodies were acquired from Invitrogen and used in a dilution 1:400.

**Behavioral testing.** Behavioral testing was conducted with both male and female mice. Animal's survival and well-being were monitored daily starting with the 1st day after TMX administration (Day 1) for at least 1 month (Day 30). $CamKIIa$-$CreERT2::Crebbp^{f/f}::Ep300^{f/f}$, $CamKIIa$-$CreERT2::Crebbp^{f/f}::Ep300^{f/+}$, and $Cam$-$KIIa$-$CreERT2::Crebbp^{f/+}::Ep300^{f/f}$ mice were first examined in the SHIRPA test[71] 2 days before TMX treatment and again 3 days after the last TMX injection (Day 12). SHIRPA phenotyping categories and scoring can be found in Supplementary Table 1.

**Stereotaxic surgeries and virus administration.** Three-month-old $Crebbp^{f/f}::Ep300^{f/f}$ mice were deeply anesthetized with a mixture of midazolam (5 mg/kg), medetomidine (1 mg/kg), and fentanyl (0.05 mg/kg) mixed in NaCl (0.9%). As soon as a total loss of reflexes was observed, the animals were positioned in a digital stereotaxic frame (Stoelting). At this point the body temperature was constantly monitored and maintained during the surgery at 37 °C using an electric blanket. Local anesthetic (EMLA 25%, lidocaine/prilocaine, AstraZeneca) was applied on the ear bars and the ophthalmologic gel (Viscotears, Bausch+Lomb) was administered on the eyes to avoid the formation of ulcers. Once the cranium was exposed, a hole was drilled in the calvaria in the location corresponding to the hippocampus. A glass capillary (Word Precision Instruments) containing the adeno-associated virus (AAV) was placed to reach the coordinates of the hippocampal hilus (−2.0, ±1.35, −1.95; in mm relative to bregma) and left in place for 5 min. 500 nl of either AAVs rAAV5-hSyn-GFP-Cre or rAAV5-hSyn-mCherry-Cre (Vector Core at the University of North Carolina at Chapel Hill) were delivered. The capillary was then left in place for another 5 min and withdrawn. After the surgery, anesthesia was reversed with subcutaneous atipamezole (0.02 mg/kg). Buprenorphine in food pellets (1 mg/ml) was placed in the home cages to reduce post-surgery pain and mice were monitored daily until full recovery.

**In vivo electrophysiology.** Mice were deeply anesthetized with 4% isoflurane (Isoflo®, Esteve Veterinaria S.A.) in 0.8 L/min oxygen and fixed in a stereotaxic setup (Narishige Group) over a heating pad at 37 °C. Isoflurane was kept at 1–2%, 0.8 L/min oxygen to maintain the anesthesia. After checking the lack of reflexes, mice were placed and fixed in a stereotaxic frame (Narishige Group). The skin of the head was cut, and the scalp and periosteum were separated. Two 1.8 mm Ø holes were made in the skull using a milling cutter (FST 18004-18, Fine Science Tools) attached to a cordless micro drill (58610V, Stoelting Co.) in the appropriated coordinates to introduce the electrodes. Then one bipolar stimulating electrode (10–15 kΩ, 325 µm Ø, TM53CCNON, World Precision Instruments) was introduced in the perforant pathway (from bregma, in mm: −4.3 AP, +2.5 ML, +1.4 DV, 12° angle), and one recording probe (single shank, 50 µm contact spacing, 32 channels; NeuroNexus Technologies) was targeted to the hippocampus CA1 and dentate gyrus regions (from bregma, in mm: −2 AP, +1.5 ML, −2 DV). Recording and stimulating electrodes were implanted following stereotaxic standard procedures and optimized based on the online recording to have the best quality of the signal in the dentate gyrus and CA1, especially taking into account the typical evoked potential in dentate gyrus. A custom-made Ag/AgCl wire was placed in contact with the skin and used as a ground. After optimizing the final position, the tissue was allowed to rest for 30 min before acquiring electrophysiological data. The position of the electrodes was confirmed post mortem. The stimulating electrode was connected to a pulse generator and current source (STG2004, Multichannel Systems) controlled by MC_Stimulus v3.4 (Multichannel Systems). Electrophysiological data from the recording probes were filtered (0.1–3 kHz), amplified and digitalized (20 kHz sampling rate for evoked potentials and 32 kHz for spontaneous activity recordings) and analyzed off-line using the Spike2 v6 (Cambridge Electronic Design Limited) or MATLAB R2016b (MathWorks) using the ICAofLFPs v1.02 package. Stimulating and recording protocols entailed spontaneous recordings (5 min) and evoked potentials, that consisted of a classical input–output (IO) stimulation protocol (stimulation intensities of 0.05, 0.1, 0.2, 0.4, 0.6, 0.8, 1, and 1.2 mA). For evaluating the excitatory post-synaptic potential the deepest slope of the evoked potential in molecular layer (dentate gyrus) was measured. To reflect the population spike (PS) the amplitude of the spike recorded in hilus was measured. Data were averaged by animal, per stimulation intensity, and then by group. Spontaneous activity signals coming from representative channels in dentate gyrus were selected to analyze the power of the frequency bands and the wavelet spectrum. Briefly, after down-sampling of spontaneous recordings to 2.5 kHz, the signals were filtered (high pass at 0.5 Hz and notch at 50 and 100) and then analyzed to extract: (a) its power density by frequency bands; (b) the wavelet spectrum, using the Fourier transformation or the wavelet spectrum analysis, respectively, implemented in the MATLAB package ICAofLFPs. Single unit activity (SUA), as local activity reflex, was analyzed using a supervised tool integrated in Spike2 software. Briefly, after applying a band pass (0.3–3 kHz, Butterworth digital filter), the different waveforms were extracted with intensity threshold set at $\pm 3^{-4}$ mV (to avoid noise) for all the putative SUA-spikes recorded nearby the recording electrode in the dentate gyrus or in CA1. After the complete scan of the electrophysiological signal, per area, we manually chose only those waveforms that clearly fit with the typical one reflecting neuronal activity (supervised procedure). The number of spikes was averaged by area, animal, and then by group to avoid an overestimation of the total $n$ for the statistical comparisons.

**Histology and image processing**. Mice were anesthetized using a mix of xylazine (Xilagesic, CALIER) and ketamine (Imalgene, MERIAL LABORATORIOS) and perfused transcardially, first with phosphate-buffered saline (PBS, pH 7.4) to removed whole blood, then with a solution containing 4% paraformaldehyde in PBS. Brains were postfixed overnight (4% paraformaldehyde) and subsequently cut on vibratome into 50 μm sections. Sections were used for immunohistochemistry (fluorescent and diaminobenzidine) or Nissl staining[65]. Some antibodies used for the fluorescent immunohistochemistry required a 30 min antigen retrieval in 80 °C sodium citrate buffer (10 mM sodium citrate, 0.05% Tween 20, pH 6.0). Golgi–Cox impregnation was performed using the FD Rapid GolgiStain™ Kit (FD NeuroTechnologies, Inc.). To this end, animals were anesthetized using a mix of xylazine and ketamine and sacrificed through cervical dislocation. The brains were instantly removed from the skull, rinsed very briefly with double distilled water, and placed in the impregnation solution at room temperature. The solution was changed to a fresh one after the first 24 h. After 10 days the brains were immersed in solution C for 72 h, after which it was changed to a fresh solution C for further 24 h. Subsequently, the brains were cut into 100 μm sections in a 1:1 mix of PBS and solution C using a vibratome. These sections were mounted on gelatin-coated slides, revealed using solutions D and E, dehydrated using ethanol and xylene and covered using Neo-Mount® (Merck). This protocol resulted in sparsely marked neurons. Brain slices labeled using Golgi staining were visualized under a bright field microscope with a motorized stage. Dentate gyrus granule neurons with intact dendritic trees were traced and reconstructed using the Neurolucida software (MBF Bioscience). These reconstructions were used to perform a Sholl analysis where the number of intersections was calculated every 2 μm starting from 5 μm from the soma. The thickness of CA1 sub-regions was quantified based on bright field images of Nissl stained sagittal brain slices, using the ×2.5 objective. Four lines were drawn perpendicularly to the CA1 and the thickness of *stratum pyramidale* and *stratum radiatum* were measured along these lines using Fiji-ImageJ software. For electron microscopy (EM) experiments, mice were anesthetized and perfused as indicated above with the addition of 2.5% glutaraldehyde in the fixation solution. Brains were cut in 100 μm-thick slices using a vibratome. Slices with dorsal hippocampus were postfixed with 1% osmium tetroxide for 1 h at room temperature. Dehydration was performed by incubating the slices in increasing ethanol concentrations and in pure propylene oxide. During dehydration, tissue was stained with 1% uranyl acetate in 70% ethanol. Slices were then embedded in the Epon resin between two Aclar sheets. After polymerization, Cornu Ammonis (CA1), *stratum radiatum*, and dentate gyrus regions fragments were cut out and stuck to an empty block of resin. Next, 75 nm sections were prepared and post stained with uranyl acetate and Reynold's lead citrate. Electron micrographs were taken with JEM 1400 transmission electron microscope at 80 kV (JEOL Ltd. 2008). Quantification of synapses and heterochromatin clumps number were performed using Fiji-ImageJ software with the *Cell Counter* plugin or macro based on the *Analyze Particles* function, respectively.

**Fluorescence-activated nuclear sorting (FANS)**. Experimental mice were sacrificed by cervical dislocation and the hippocampal tissue was extracted from the brains. This tissue was subsequently homogenized using a Douncer tissue grinder (Kontes® 2 ml) in a buffer containing 0.5% IGEPAL and filtered on a 35 μm nylon mesh (Falcon #352235). The resulting suspension of hippocampal nuclei was stained against NeuN and DAPI. At this point three independent samples were pooled to form a single replicate that was centrifuged in Optiprep (MERCK) gradient. Nuclei purified this way were sorted using a BD FACS Aria III Flow Cytometer based on their size (FSC), complexity (SSC), and DAPI and NeuN signals. Specific gate was set up in the flow cytometer to isolate nuclei weakly immunofluorescent for NeuN in dKAT3-ifKO (but still expressing higher NeuN levels than non-neuronal cell-types) for the ATAC-seq experiments (NeuN+, Fig. 4c). Approximately 50,000 nuclei were obtained in each sample (41.5% of events in control and 46.45% in dKAT3-ifKO).

**Genomic data processing and access**. Adapters were trimmed using cutadapt v1.18 and aligned to mm10. Only reads longer than 25 bp, with mapq > 30 and mapping to nuclear chromosomes were used in ulterior analyses. Data was processed with custom R scripts (R version 3.5.1, 2018), Samtools v1.9, Bedtools v2.27.1, and DeepTools v3.2.0. Whole genome alignments were normalized to 10× RPM (read per 10 million sequenced reads) and visualized using IGV v2.5.0.

mRNA-seq: Extraction of RNA from the hippocampal tissue was performed using TRI-reagent (MERCK)[72]. Total RNA was treated with DNase I (Qiagen) and its quality was confirmed using nanodrop, Bioanalyzer and RT-qPCR assays. Three independent samples were prepared for control and dKAT3-ifKO mice. Each sample, corresponding to a single mouse, was used to prepare a polyA library and sequenced on a HiSeq 2500 sequencer (Illumina, Inc.). Reads were aligned with HISAT2 v2.1.0 to the mouse genome (mm10). Mapped reads were annotated to genes from Ensemble (GRCm38.89) and quantified using HTseq v0.11.1. Differential expression analysis was performed using DESeq2 v1.10.0[73]. Genes with FDR < 0.05 and |log2FC| > 1 were considered significantly deregulated; the number of DEGs would be much higher if we used a lower log2FC threshold. GO terms were analyzed using GOstats v2.44.0. Tissue-specific gene expression was taken from GTEx[74], selecting *Brain-Hippocampus*, *Heart-Left Ventricle*, *Liver*, and *Lung*.

Assay for the transposase accessible chromatin followed by high-throughput sequencing (ATAC-seq): Sorted neuronal nuclei were centrifuged and resuspended

in the transposase reaction mix (TD buffer and Tn5 transposase, Illumina). The mix was placed in 37 °C for 30 min and immediately after the DNA was extracted using Qiagen MinElute PCR Purification Kit. A DNA library was then prepared using Custom Nextera PCR primers 1 and 2[22,75]. We monitored the saturation of the library using an RT-PCR and afterwards extracted the DNA using the kit. DNA libraries were sequenced using HiSeq 2500 sequencer (Illumina, Inc.). Paired-end reads were aligned with Bowtie2 v2.3.4.2 to mm10 mouse genome. Duplicated reads were removed with Picardtools v2.18.21 (https://broadinstitute.github.io/picard/). Only paired reads were used for posterior analysis. Peak calling was performed with MACS2 v2.1.1. Following ENCODE recommendations, to filter the most reproducible and better quality peaks, called peaks from ChIP replicates were subjected to irreproducible discovery rate (IDR) selection and only peaks with IDR < 0.15 were taken for downstream analysis. DARs analysis was performed using DiffBind v2.6.6. Regions with FDR < 0.05 and |log2FC| > 1 were considered significantly regulated. DARs were annotated to closest genes from Ensembl (GRCm38.89) using the Bioconductor package ChIPpeakAnno v3.20.1. Predictive relationship of DARs to gene expression changes was performed using BETA v1.0.7[23] using as reference genes from Ensembl (GRCm38.89). Motif analysis of ATAC regions was performed using MEME-suite v4.12.0. For digital footprint we used the ATAC-seq dedicated software HINT v0.12.3.

ChIP-seq: Reads were aligned with Bowtie2 v2.3.4.2 to the mm10 mouse genome. Peak calling was performed with MACS2 v2.1.1. Following ENCODE recommendations, we used IDR < 0.05 to avoid false positives and retrieve the most reliable peaks. Heatmaps were performed with DeepTools v3.2.0. Circos plot was drawn with the R package Circlice v0.4.8. Differential protein binding analysis was performed using DiffBind v2.6.6. Regions with FDR < 0.05 and |log2FC| > 1 were considered significantly regulated. Annotation and analysis of ChIP peaks was performed with ChIPseeker v1.22.1 and ChIPpeakAnno v3.20.1. ChIP peaks were annotated to the closest gene from Ensembl (GRCm38.89). Motif analysis of regions occupied by ChIP peaks was performed using MEME-suite v4.12.0. For the classification of regulatory regions categorized by genomic features, neuronal and pancellular KAT3 peaks were split according to H3K27ac enrichment and H3K4me1/H3K4me3 content (information of H3K4me1 ChIP-seq from ENCODE (ENCFF545CTN) and H3K4me3 ChIP-seq from ref. [72]). Most H3K4me3-rich regions lie up to 1 kb from the TSSs corresponding to what could consider promoters of active genes, while H3K4me1-rich peaks preferentially locate into introns and intergenic regions and were labeled as enhancers. We defined as neuronal enhancers those regions that contain neuronal peaks for KAT3 binding, ATAC-seq, H3K27ac, enriched in H3K4me1 and located in introns or intergenic regions. Peaks that contain KAT3, ATAC-seq, H3K27ac, and H3K4me3 enrichment located at promoters were labeled as active promoters. To retrieve putative neuronal super-enhancers we used similar criteria than Whyte et al.[25], H3K27ac regions closer than 5 kb were stitched together and tested according to the afore referred criteria. Regions containing neuronal enhancers were carried over for further analyses. Those regions longer than 5 kb (the length at which gene expression deviates from linear correlation in Fig. 5e) and associated with highly expressed genes (i.e., expression 1 order of magnitude higher than average, 100 RPKM) were labeled as neuronal super-enhancers (SEnh). The information of Neurod2 ChIP-seq was extracted from ref. [29]. The Phenotype analysis was performed using the WEB-based application GEne SeT AnaLysis Toolkit (http://www.webgestalt.org/). The p300 ChIP-seq data for heart (ENCSR777VNA), liver (ENCSR765RPR) and lung (ENCSR527DME) of C57BL/6 P0 mice were obtained from ENCODE (https://genome.ucsc.edu/ENCODE/downloadsMouse.html).

**Single-nucleus RNA sequencing and analysis**. For the single-nucleus RNA-seq experiment, dKAT3-ifKO mice were sacrificed either 2 weeks (2w) or 1 month (1m) after TMX administration. The hippocampi of each mouse were dissected in cold PBS and transferred to a Dounce homogenizer containing 1 ml of ice-cold MACS buffer (0.5% bovine serum albumin (BSA), 2 mM ethylenediaminetetraacetic acid, PBS 1×) and homogenized 12–15 times with the pestle. The cell suspension was transferred to a 2 ml tube and centrifuged 15 min at 500g and 4 °C. Cell pellet was resuspended in 2 ml of lysis buffer (10 mM Tris–HCl, 10 mM NaCl, 3 mM MgCl$_2$, 0,1% IGEPAL) and kept 5 min on ice. Samples were then spun down at 500g for 30 min in a pre-chilled centrifuge. The pellet was resuspended in PBS 1×, 1% BSA, and sorted in a BD FACS Aria III. 15,000 nuclei per sample (pool of 2 animals) were loaded into the single cell A Chip and then the generation of barcode-containing partitions was carried out with the Chromium Controller (10X Genomics). Chromium Single Cell 3′ Library & Gel Bead Kit v2 was employed for post-GEM-RT clean-up, cDNA amplification and the generation of barcoded (ChromiumTM i7 Multiplex Kit) libraries. Libraries were sequenced to an average depth of 290–310 million reads per sample on an Illumina HiSeq 2500 sequencer. Quality control of sequenced reads was performed using FastQC v0.11.9. Sequenced samples were processed using the Cell Ranger v2.2.0 pipeline (10X Genomics) and aligned to the CRGm38 (mm10) mouse reference genome customized to count reads in exons and introns (pre-mRNA) (gene annotation version 94). We retrieved 1791 (control), 1133 (dKAT3-ifKO 2w), and 1465 (dKAT3-ifKO 1m) high quality nuclei per sample. Mean reads per nucleus were 172,879 (control), 271,535 (dKAT3-ifKO 2w), and 197,840 (dKAT3-ifKO 1m). Single-nucleus RNA-seq data were subsequently pre-processed and further analyzed in R using Seurat v2.3.4. Filtering parameters were as follows: genes, nCell <5; cells, nGene <200. Data were then normalized using global-scaling normalization (method:

LogNormalize, scale.factor = 10.000). To identify major cell populations in the dorsal hippocampus of adult mice, control and dKAT3-ifKO datasets were analyzed separately. Highly variable genes (HVGs) were detected using *FindVariableGenes* function with default parameters. Then, normalized counts on HVGs were scaled and centered using *ScaleData* function with default parameters. Principal component analysis (PCA) was performed over the first ranked 1000 HVGs, and cluster detection was carried out with Louvain algorithm using 20 first PCA dimensions at resolution = 0.6 (the default and the optimal according to cell number, data dispersion, and co-expression of previously reported cell markers). Visualization and embedding were performed using tSNE and UMAP over PCA using the 20 first PCA dimensions. UMAP plots of gene expression show normalized count (UMIs) per nucleus. The equalized expression between fixed percentiles was plotted according to the following criteria: the minimum expression was adjusted to 25% and the maximum expression was adjusted to 95% in all expression plots. For longitudinal analysis, datasets from the 3 conditions were merged and HVGs were identified for each dataset as indicated above. Only HVGs that were detected in all datasets were used to perform the visualization and embedding. Clustering was performed on merged dataset from 3 conditions and populations were identified combining these results with clustering information obtained in control and dKAT3-ifKO-1m datasets separately, together with co-expression of population markers. Differential expression analysis (DEA) was used to identify population gene markers. For DEA, the nuclei of each population were contrasted against all the other nuclei using the merged dataset using Wilcoxon Rank Sum test on normalized counts. Trajectory analysis was performed in unbiased manner using Monocle 2 (v.2.8.0)[21]. First, we selected from Seurat merged dataset those clusters containing principal excitatory neurons and transformed this subset to a Monocle object. The following population clusters were present in the Monocle object: Granule cell, CA3 pyr. neuron, CA1 pyr. neuron A, CA1 pyr. neuron B, Former CA pyr. neuron, and Former granule cell. We then filtered those genes whose expression was detected in less than 10 cells per cluster. 13,935 genes were retained after this filtering step. Next, the size factor and dispersion of the subset was estimated, and data was normalized and preprocessed. Genes under the minimum level detection threshold of 0.1 (average expression level) were further removed leaving 2406 genes to perform discriminative dimensionality reduction with trees (DDRTree). DDRTree was applied inside the function reduceDimension, with default parameters: norm_method = "log", pseudo_expr = 1, relative_expr = TRUE, auto_param_selection = TRUE (automatically calculate the proper value for the ncenter (number of centroids)) and scaling = TRUE (scale each gene before running trajectory reconstruction). Prior the dimensional reduction, the function *reduceDimension* also performed a variance-stabilization of the data (because the expressionFamily of the data was negbinomial. size). Finally, the cells were ordered according to pseudo-time with the function *orderCells*, which added a pseudo-time value and state for each cell.

**Statistical analysis**. All the statistics in the following work have been done using RStudio v3.5.1, GraphPad Prism v8.0.1, and MATLAB R2016b, depending on the experiment. Statistical tests used in the study are indicated in the figure legend. All statistical tests used in this study were two-sided, except for the Hypergeometric test. For a comparison of two groups, each group was first tested for the normality using Shapiro–Wilks test. If normality assumption was not violated, a *t*-test was performed. If the normality null hypothesis was rejected, Mann–Whitney U test was performed. In the analysis of the SHIRPA paradigm, Fisher exact test was used for the categories carrying just two possible outcomes (indicated in Supplementary Table 1). For multiple testing of the same sample, *p*-values were corrected using Bonferroni method. For two factor comparisons, two-way ANOVA was used. In all bar plots, the height represents the mean and the error bars the standard error of mean (SEM).

**Reporting summary**. Further information on research design is available in the Nature Research Reporting Summary linked to this article.

## Data availability
The genomic data sets generated in this study can be accessed at the GEO public repository using the accession number GSE133018. We used the following publicly available webtools: GEne SeT AnaLysis Toolkit (http://www.webgestalt.org/) and MEME-suite (MEME-suite.org/). The p300 ChIP-seq data for heart (ENCSR777VNA), liver (ENCSR765RPR), and lung (ENCSR527DME) of C57BL/6 P0 mice were obtained from ENCODE (https://genome.ucsc.edu/ENCODE/downloadsMouse.html).

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

## Acknowledgements

The authors thank P. Arlotta, O. Hobert, N. Flames, E. Herrera, M.A. Nieto, A. Rada-Iglesias, and J.V. Sanchez-Mut for critical reading of the manuscript. The authors thank A. Caler, N. Cascales-Picó, M. Llinares, A. Medrano-Fernández, and S. Rivero for their assistance in specific experiments and V. Makarov for the ICAofLFPs MatLab package. M.L. is recipient of a Santiago Grisolia fellowship given by the Generalitat Valenciana, J. M.C. is recipient of a fellowship from the Spanish Ministry of Education, Culture and Sport (MECD), J.F.-A. and C.M.N. are recipients of fellowships from the Spanish Ministry of Science and Innovation (MICINN). The ultrastructure research was supported by the Polish National Science Center Grant UMO-2014/15/N/NZ3/04468 and by the European Regional Development Fund POIG 01.01.02-00-008/08. J.P.L.-A. research is supported by Grants RYC-2015-18056 and RTI2018-102260-B-I00 from MICINN cofinanced by ERDF. A.B. research is supported by Grants SAF2017-87928-R, PCIN-2015-192-C02-01, and SEV-2017-0723 from MICINN co-financed by ERDF, PROMETEO/2016/026 from the Generalitat Valenciana, and RGP0039/2017 from the Human Frontiers Science Program Organization (HFSPO). The Instituto de Neurociencias is a "Centre of Excellence Severo Ochoa".

## Author contributions

Conceptualization: M.L., R.M.-V., B.d.B., and A.B. Methodology: M.L., B.d.B., J.F.-A., C. M.N., R.O., A.A.S., and J.M.C. Software: R.M.-V. and A.M.-G. Investigation: M.L., B.d.B., and J.M.-R. Data Curation and Visualization: R.M.-V., A.M.-G., and M.L. Writing—Original Draft: A.B. and M.L. Supervision: A.B., J.P.L.-A., S.C., and G.M.W. Funding Acquisition: A.B.

## Competing interests

The authors declare no competing interests.
