## [Peer Review File · Nature Communications]

Reviewers' Comments:

Reviewer #1:

Remarks to the Author:

In this manuscript the authors examine the neuronal consequences of knocking out both members of the KAT3 family of acetyltransferases in differentiated Camk2a+ neurons of the CNS. Similar to what has been shown for germline deletion, the authors find that the two genes are mostly functionally redundant, because both alleles of both CBP and p300 must be knocked out to observe a significant phenotype. However, when both genes are gone the phenotype is severe, resulting in rapid death in most of the mice. They augment their in vivo studies with local hippocampal knockout and primary culture studies to examine neuronal phenotypes with prolonged deletion.

At the cellular level the authors found no evidence for cell death up to two months after the knockout, but they did see substantial loss of neuropil suggesting that the neurons shrink and lose their synaptic connections, which was supported by data. At the level of RNA expression, they see substantial loss of many neuronal genes. Genes that are downregulated upon loss of KAT3s tend to be near KAT3 ChIP peaks in enhancers that are lost in the double knockout. Consistent with the role of KAT3s as HATs, there is a global reduction in H3K27ac in the double knockout neurons that is pronounced at neuronal enhancers and superenhancers. Neuronal bHLH binding sites are enriched in the KAT3 target sites, and bHLH factors including Neurod2 are among the downregulated genes, however enhanced expression of Neurod2 was not sufficient to rescue dKAT3 KO. By contrast dCas9-p300 mediated local rescue of H3K27ac at a single gene promoter was sufficient to rescue expression of a target gene.

This is an impressive piece of work and the data are of high quality. This paper will provide a novel addition to the literature on gene regulation in the nervous system because it shows data that are foundational for understanding the functions of this important chromatin regulator in differentiated neurons.

I have only one significant concern and it has to do with the interpretation of the data. Specifically, the authors imply the claim that the function of the KAT3 family in differentiated neurons is specific for the maintenance of cell fate. Furthermore, they suggest that KAT3s have a selective function in maintaining chromatin state at enhancers versus promoters. However, I am not convinced the authors can make these conclusions based solely on their current analyses.

At the RNA level they suggest that KAT3s are specific for neuronal genes because they see a selective effect on neuronal-specific genes versus pancellular genes in their RNAseq. However they are sequencing a mixed population of cells, only a subset of which lack the KAT3s, and the effects on pancellular genes in the neuronal fraction could be masked. It matters significantly to understand if the transcription of pancellular genes might also be impaired in the knockout neurons. This is important both for the conceptual conclusions of the paper, and also for interpretation of the sequencing data, which will incorporate total reads in the significance calculations. Thus if there were a global reduction in RNA in the knockout neurons, this could confound the statistical analysis of the RNAseq data which usually output measures of the expression of single genes relative to total read depth.

The authors may already have data to address this concern perhaps in their single cell sequencing data (though that method is poorly quantitative) and certainly it could be in their culture data, both of which do not have the mixed population confound. The authors could also do simple experiments that are not confounded by the quantitative biases of overall RNA levels like single-molecule RNA FISH to quantify the expression pancellular genes on the brain sections from Figure 2F (or the cultures in 2H) or they could add quantitative immunostaining for pancellular proteins (in addition to neuronal proteins) to the images in Figure S6C. They could also provide a quantitative measure of total mRNA per cell in the double knockout neurons to determine whether

the reduction in transcription is global versus specific.

If it were the case that all (or even most) gene expression in the knockout neurons were impaired, this would still be equally important for understanding the functions of the KAT3 family in differentiated cells, but it would require changing the text of the paper to back off the “fate maintenance” angle. An example from the literature for the authors to consider as a model for addressing this concern is the work from the Jaenisch lab on MeCP2, which showed a global decrease in transcription in MeCP2 mutant neurons (PMID: 24094325).

At the chromatin level, again the mixed population of cells confounds the promoter/enhancer analysis at a global level. The data shown at least in the main text figures compare enhancers of neuronal genes to promoters of pan-cellular genes. However the authors could look at the promoters of neuronal-specific genes and compare them to enhancers of those genes to ask if there was a selective effect at one kind of regulatory element or the other. This may be buried within the supplementary figures but some specific examples pulled out to see would be helpful given the density of the data in this manuscript and the importance of this comparison for the narrative the authors have chosen to tell. The binding and histone regulation is exceptionally important to be convinced of given the argument the authors make in the discussion about the differential localization of other acetyltransferases.

Minor concerns:

- 1) Are the cell bodies shrinking? That would be consistent with the idea the cells are just slowly running down.
- 2) The single nucleus data could be described better. What is a “nonfunctional interstate deadlock”? Do the knockout neurons still express enzymes needed to survive? Do they express anything? Or are there simply no transcripts found in these cells? Are they more likely to fail QC, which would be expected if they simply have very little transcription?
- 3) It is confusing that in the ChIP the strongest binding sites for both CBP and p300 fail to go away in the double knockout (Fig. 4A). This must be coming from the non-neuronal cells, but it is still confusing.
- 4) The histone immunostainings in 5a and S10a should be quantified. I appreciate that another paper recently reported on a similar specificity of KAT3s for K27 over K9/14 on histone H3 but a strong quantification of these data here would be nice to see in the literature.
- 5) One final comment to the authors to take as they may. This is an impressive paper with beautiful data. But there is honestly too much of it and sometimes it distracts from the ability to follow the story without adding too much. For example, the data in Figure S10g-h, which is derived from the datasets authors’ recent Nat Neurosci paper. This data is not sufficiently explained for the reader of this paper to fully understand it, and it isn’t used to make a strong point here. There are a number of data presentations in the supplement that I similarly felt were not necessary for the story, or if they were necessary then they were not sufficiently described in the text for me to be able to understand. In either case, I want to assert that this story is so strong that it does not need any extra data and as mentioned above in the section about the acetylation differences at enhancer and promoters, including too many other less important analyses may well be hiding data that supports the main points.

Reviewer #2:

Remarks to the Author:

Dear editorial team, dear authors,

Please find below my comments on the Manuscript by Lipinski and colleagues.

Manuscript #: NCOMMS-19-37453-T

1- General comments

The manuscript "Neuronal identity is maintained in the adult brain through KAT3-dependent enhancer acetylation" by Lipinski and colleagues explores the role of two key acetyltransferases, (CBP and p300) in cell-fate maintenance. Using inducible knock-out animals, they show that the ablation of both genes leads to loss of specific neuronal identity, exemplified by the loss of electrical activity, proper connections and aberrant dendrite morphologies. The authors describe in genome-wide assays the consequence of the loss of function. Using a combination of transcriptomics and chromatin accessibility approaches, they document the specificity of action of CBP and p300, and propose some of the mechanisms by which these chromatin remodelers can lock specific neuronal populations and astrocytes in their respective identities.

Overall this manuscript is a well-written documentation combining in vivo and in vitro experimental series to demonstrate the role of KAT3 in the adult brain. It extends the field by detailing the specificity of acetylation marks on neuronal genes to maintain cell-fate. Altogether their findings are convincing and the questions being addressed are likely to further influence the way we will develop ideas in the field. Furthermore, the experimental design of the study is robust and in general the data appears technically rigorous.

2- Specific comments are listed below:

-The central point that is made by the authors regards the loss of neuronal identity. While it is clear that neuronal function is altered (activity, synapses, dendritic morphologies), it is still uncertain how non-neuronal the cells become. The conserved proportion of NeuN positive cells in the hippocampi from dKAT3-fKO suggest that the neuronal state is still maintained. Also, the conclusion that they acquire a "molecularly undefined fate" sounds elusive. It would be informative to use the transcriptomic data to more precisely define the default cell type or cell state that the mutant cells adopt. Stating that they do not express any markers is not sufficient. At least the authors should report more clearly in the text or the main figures the main genes that are specifically expressed in these so-called non-neuronal population.

-While the text is well written, wording is often vague. I understand but it would be easier to grasp the details with a more precise description of the results in the text (See eg. l. 114, 115, 120, 121, 123, 127, 128, 150, 193, 258...).

- The use of an inducible system to ablate specifically the KAT proteins in the forebrain is a suitable way to address the question. However, the Cre used here (Camk2a-Cre/ERT2) has been reported to already demonstrate activity even in the absence of tamoxifen (Madisen et al., 2010; doi:10.1038/nn.2467). Did the authors take into account the sparse expression of the Cre prior the induction? Would it be possible to mention it in the text or discussion?

-The description of the phenotype (second paragraph of the results section, l.117-122) indicates that there is a failure of the mice to hydrate/feed properly. How much of the hydration problem still persists when the animals obtained facilitated access to food/water? Have you performed any test to control for the restoration of proper hydration and weight?

-Figure 1d-e: How confident can we be in the phenotype observed when the t-test is performed with n of 2 in the mutant condition (Fig. 1d)? What about the ANOVA (Fig. 1e) with two small groups as well (n=2 after 2 months, n=1 after 7 months). These low "n" values may somewhat impede the power of the analysis of these important phenotypes. How assertively can we interpret

these results?

-The OB in the mutant looks very large. Is that often observed? Is the Cre expressed here as well?

- The single-cell transcriptomes are shown using a UMAP, which give a good overview of the molecular diversity. It also reveals an apparent regression in cell-fate specification. However, since the trend indicates a loss of differentiation, to better emphasize the shift toward undefined cell states I would recommend a trajectory-based representation/organization. Organizing the cells across a pseudotemporal alignment may help discriminate between a change in cell type and a regression along a differentiation axis.

-The authors report a slight change in heterochromatin structures (Fig S3c-d). How does this phenotype relate to the absence of H3K9me3 levels reported? (See Result section; l. 271-273)

-The rescue experiment is a very interesting attempt to further explore the mechanistic specificity of the KAT phenotypes. This rescue being conducted on Neurod2 based on a severe downregulation, how does this downregulation compare to the effects on other genes? Is there any other reason to select that specific TF? It would have been helpful to evaluate how negative control genes behave upon such CRISPR modulation.

-It is not clear to me how the one day between "gRNA Cherry transfection + dCas9KAT" and the analysis is enough to elicit the described effect.

- In describing the Movie M1 the authors state that "Shortly afterwards, the same mice showed severe ataxia, and loss of the righting reflex, escaping response and tail-suspension-evoked stretching". However, there is no clear quantification of these effects. The behavior there could be explained by sleep or food disorder, or by a sporadic sickness.

3- Minor points:

-The first words of the abstract tend to diminish the work from other groups. "Very" is not necessary.

-The study is based on experiments mostly in hippocampal neurons and astrocytes. This should be explicitly stated in the abstract.

-Line 104: "while brain regions in which the CaMKII α promoter is not active, such as the cerebellum and the basal ganglia, were spared." Could you add a reference to this statement?

-Line 119: Is the data related to the statement shown? ("all mice died within the first 2-3 weeks after TMX administration")

-Line 260: How enhancers were defined? A reference? By the presence of p300? Chromatin landmarks?

-Fig. 8c and 8f: The scale bars are missing.

-Fig S3c: What was the statistical test performed here?

-Fig. S3g: any quantification for the Cas3 signals?

-Fig. S6c: "staining against ...genes NeuroD2": is the gene stained or is it the protein? Why genes is plural?

-Fig. S7: In the caption title it is RNA-seq (not RNA-seg).

-Fig. S10c: H3K9me3 seems reduced in the image shown. Same in panel S10a. Did the authors measured/quantify the levels of this histone mark?

-Fig. S10g: do you mean one-tailed Mann-Whitney test?

Best regards,

Pierre Fabre

Point-by-point response to Reviewers

We thank the Editor and Reviewers for their thoughtful and constructive comments. We conducted new analyses and experiments according to the Reviewers comments. These new results are presented in new panels in Figures 2, 3, S3 and S11 (former Fig. S10), and in the new Figure S8. In addition, we have revised the text and several other figures to correct a few minor mistakes and incorporate the changes suggested by the Reviewers. We believe that with these changes and additions we have effectively addressed the Reviewers' comments and the manuscript is significantly improved. Our point-by-point response to all the issues identified during the review process is below.

Reviewer #1 (Remarks to the Author):

In this manuscript the authors examine the neuronal consequences of knocking out both members of the KAT3 family of acetyltransferases in differentiated Camk2a+ neurons of the CNS. Similar to what has been shown for germline deletion, the authors find that the two genes are mostly functionally redundant, because both alleles of both CBP and p300 must be knocked out to observe a significant phenotype. However, when both genes are gone the phenotype is severe, resulting in rapid death in most of the mice. They augment their in vivo studies with local hippocampal knockout and primary culture studies to examine neuronal phenotypes with prolonged deletion.

At the cellular level the authors found no evidence for cell death up to two months after the knockout, but they did see substantial loss of neuropil suggesting that the neurons shrink and lose their synaptic connections, which was supported by data. At the level of RNA expression, they see substantial loss of many neuronal genes. Genes that are downregulated upon loss of KAT3s tend to be near KAT3 ChIP peaks in enhancers that are lost in the double knockout. Consistent with the role of KAT3s as HATs, there is a global reduction in H3K27ac in the double knockout neurons that is pronounced at neuronal enhancers and superenhancers. Neuronal bHLH binding sites are enriched in the KAT3 target sites, and bHLH factors including Neurod2 are among the downregulated genes, however enhanced expression of Neurod2 was not sufficient to rescue dKAT3 KO. By contrast dCas9-p300 mediated local rescue of H3K27ac at a single gene promoter was sufficient to rescue expression of a target gene.

This is an impressive piece of work and the data are of high quality. This paper will provide a novel addition to the literature on gene regulation in the nervous system because it shows data that are foundational for understanding the functions of this important chromatin regulator in differentiated neurons.

We thank the Reviewer for the very positive appreciation of our work. We also believe that this work represents an essential contribution towards understanding the role of KAT proteins and the molecular mechanisms underlying the maintenance of cell identity in postmitotic cells.

I have only one significant concern and it has to do with the interpretation of the data. Specifically, the authors imply the claim that the function of the KAT3 family in differentiated neurons is specific for the maintenance of cell fate. Furthermore, they suggest that KAT3s have a selective function in maintaining chromatin state at enhancers versus promoters. However, I am not convinced the authors can make these conclusions based solely on their current analyses.

We understand the concerns of Reviewer #1 and revised the text to avoid any overstatement. We do not propose that the only function of KAT3 proteins in differentiated neurons is the maintenance of cell fate; these proteins also play a role as KAT and transcriptional co-activator that is likely essential in neuronal plasticity processes and supporting activity-driven

transcription, however these other roles (described in the case of CBP in previous and ongoing studies of our lab) are occluded by the dramatic and rapid phenotype triggered by the combined ablation of CBP and p300. We have rephrased some sentences to clarify this view. We also revised the text to indicate that KAT3 proteins play a role in maintaining chromatin state at enhancers and promoters (see also our detailed response below and the new Figs. S11e-f). Our results suggest that the role of CBP and p300 in the promoters of pancellular genes could be compensated by other KATs, while their role in enhancers and promoters of neuron-specific genes seems to be more difficult to compensate. We revised the title of the manuscript and the abstract to avoid any overstatement.

At the RNA level they suggest that KAT3s are specific for neuronal genes because they see a selective effect on neuronal-specific genes versus pancellular genes in their RNAseq. However they are sequencing a mixed population of cells, only a subset of which lack the KAT3s, and the effects on pancellular genes in the neuronal fraction could be masked. It matters significantly to understand if the transcription of pancellular genes might also be impaired in the knockout neurons. This is important both for the conceptual conclusions of the paper, and also for interpretation of the sequencing data, which will incorporate total reads in the significance calculations. Thus if there were a global reduction in RNA in the knockout neurons, this could confound the statistical analysis of the RNAseq data which usually output measures of the expression of single genes relative to total read depth.

We thank the Reviewer for bringing out this important point. As indicated, we cannot fully discard that some pancellular genes may be also affected. Given the broad presence of CBP and p300 genes throughout the genome, thousands or tens of thousands genes are potentially affected by the elimination of CBP and p300. We however believe that our data clearly show that the most affected genes are the neuron-specific genes. Still, cellular heterogeneity is a relevant confounding factor in the interpretation of transcriptome data. The results of neuron-specific ATAC-seq screen strongly support the larger sensitivity of neuronal-specific genes to double KAT3 ablation in neurons compared to pancellular genes. To further clarify this point, we conducted the analysis with the single nucleus RNA-seq data suggested by the Reviewer (new Figure S8). The results of these additional analyses also supported a gene set specific effect rather than a global effect (see also our response to the next point).

The authors may already have data to address this concern perhaps in their single cell sequencing data (though that method is poorly quantitative) and certainly it could be in their culture data, both of which do not have the mixed population confound. The authors could also do simple experiments that are not confounded by the quantitative biases of overall RNA levels like single-molecule RNA FISH to quantify the expression pancellular genes on the brain sections from Figure 2F (or the cultures in 2H) or they could add quantitative immunostaining for pancellular proteins (in addition to neuronal proteins) to the images in Figure S6C. They could also provide a quantitative measure of total mRNA per cell in the double knockout neurons to determine whether the reduction in transcription is global versus specific.

We thank the Reviewer for this great suggestion. We conducted the analysis with the single nucleus RNA-seq data suggested by the Reviewer (new Figure S8). Briefly, we show the raw total unique transcripts (raw UMI) per cell as well as the total number of detected genes per cell across the different populations in violin plots. First, we show violin plots for all identified populations (panel S8a). These plots (raw UMI and nGene) show that our experiment detects cell populations with ample differences in RNA content (e.g., neuronal cells show much higher UMI content per cell and nGene detection per cell than microglia). This ample dynamic range should allow the clear detection of an eventual drop in global

transcriptional levels for a given population within in the datasets. However, our data show that the levels of total UMI per cell (panel S8b) and total detected genes per cell (panel S8c) in “former neuron” is in the range of that observed in intact excitatory neurons. Note that cells were sequenced to comparable read depths with the “dKAT3-KO 1-month” group having a higher mean as compared to control group (Control mean = 172,891; dKAT3-KO 1-month = 197,840). However, the median UMI count and gene detection rates were slightly lower in the “dKAT3-KO 1 month” as compared to control (median UMI count: control = 1,092; dKAT3-KO 1-month = 857. Median genes per cell: control = 749; dKAT3-KO 1-month = 656) which may represent a decrease in the complexity of the library due to the selective effect on the transcriptome of postmitotic neurons lacking CBP and p300 KATs. Taken together, these results argue against a global reduction in transcriptional levels upon combined ablation of CBP and p300. For comparison, we added to these figures two cell populations (interneurons and oligodendrocytes) that show a large number of detected nuclei in our datasets and are in different levels of detection for these two parameters.

Regarding the second suggestion, we would like to point out that the results for primary neuronal cultures (PNC) presented in Figure 2i (RT-qPCR in PNC) already suggested that the impact is exclusive of neuronal genes (we used *Gapdh* as a reference). We have now conducted some additional RT-qPCR assays in PNCs that further confirm the specificity of the effect (new Figure 2i).

If it were the case that all (or even most) gene expression in the knockout neurons were impaired, this would still be equally important for understanding the functions of the KAT3 family in differentiated cells, but it would require changing the text of the paper to back off the “fate maintenance” angle. An example from the literature for the authors to consider as a model for addressing this concern is the work from the Jaenisch lab on MeCP2, which showed a global decrease in transcription in MeCP2 mutant neurons (PMID: 24094325).

We hope that the additional experiments, analyses and the revised text had effectively addressed the Reviewer’s concerns. Although, obviously we cannot discard a more general and subtler effect, we believe that our experiments consistently show that some genes are more sensitive than others to the lack of KAT3 proteins and that this set of highly sensitive genes in the case of excitatory neurons corresponds to the genes that are commonly associated with their neuronal identity.

At the chromatin level, again the mixed population of cells confounds the promoter/enhancer analysis at a global level. The data shown at least in the main text figures compare enhancers of neuronal genes to promoters of pancellular genes. However the authors could look at the promoters of neuronal-specific genes and compare them to enhancers of those genes to ask if there was a selective effect at one kind of regulatory element or the other. This may be buried within the supplementary figures but some specific examples pulled out to see would be helpful given the density of the data in this manuscript and the importance of this comparison for the narrative the authors have chosen to tell. The binding and histone regulation is exceptionally important to be convinced of given the argument the authors make in the discussion about the differential localization of other acetyltransferases.

We thank the Reviewer for this suggestion. We conducted a new analysis comparing the effect of combined CBP and p300 loss in the promoter and enhancers of neuronal-specific genes (new Figures S11e and S11f). Our results show that occupancy is severely affected both in enhancers and promoters, although the changes are more abundant and significant in the case of enhancers. According to the Reviewer’s suggestion, we presented some examples in Figures 5l and 8a.

Minor concerns:

1) Are the cell bodies shrinking? That would be consistent with the idea the cells are just slowly running down.

We do not observe a significant shrinkage of the cell bodies or nuclei. We attach some confocal and electron microscopy images for Reviewer #1 perusal (Figure 1 for Reviewers).

2) The single nucleus data could be described better. What is a “nonfunctional interstate deadlock”? Do the knockout neurons still express enzymes needed to survive? Do they express anything? Or are there simply no transcripts found in these cells? Are they more likely to fail QC, which would be expected if they simply have very little transcription?

We thank the Reviewer for this comment. Maybe she/he overlooked the volcano plots presented in Figure 2b that show that several thousands of housekeeping genes (including those encoding for basic metabolism enzymes) are not differentially expressed in dKAT3-ifKOs. In addition, we have now conducted additional analyses in our single nucleus RNA-seq experiment that confirm that the cells isolated from the hippocampi of dKAT3-ifKOs still express thousands of unique transcripts (in fact, if this were not the case, they would not have been detected as cells in the initial snRNAseq analysis). The percentage of *healthy* cells, according to the criteria used in scRNA-seq analysis, was the same in the control and dKAT3-ifKO preparations. These results are presented in the new Figure S8.

3) It is confusing that in the ChIP the strongest binding sites for both CBP and p300 fail to go away in the double knockout (Fig. 4A). This must be coming from the non-neuronal cells, but it is still confusing.

We believe that this result further underscores the specificity of the effect. CBP and p300 are detected in numerous promoters, particularly in the promoters of highly expressed genes. This signal comes from both neuronal and non-neuronal chromatin. The severe loss of CBP and p300 binding in neuronal chromatin of neuronal-specific genes causes the expected redistribution of reads in the library leading to an apparent increase of CBP/p300 binding at the promoter of highly expressed ubiquitous (non neuronal-specific) genes. As indicated by the Reviewer, this signal comes from the chromatin of non-neuronal cells.

4) The histone immunostainings in 5a and S10a should be quantified. I appreciate that another paper recently reported on a similar specificity of KAT3s for K27 over K9/14 on histone H3 but a strong quantification of these data here would be nice to see in the literature.

The immunostaining of brain slices does not provide a very precise quantification of acetylation changes. We intended to present a qualitative result rather than a quantitative one, but we performed nevertheless the quantification requested by the Reviewer. We found significant differences between dKAT3-ifKOs and controls in the case of H3K27ac in Figure 5a (this information has been added to the Figure legends). However, we do not detect significant difference in fluorescence intensity in the images for H3K9/14ac in Fig S11a (former Fig. S10a).

5) One final comment to the authors to take as they may. This is an impressive paper with beautiful data. But there is honestly too much of it and sometimes it distracts from the ability to follow the story without adding too much. For example, the data in Figure S10g-h, which is derived from the datasets authors' recent Nat Neurosci paper. This data is not sufficiently explained for the reader of this paper to fully understand it, and it isn't used to make a strong point here. There are a number of data presentations in the supplement that I similarly felt

were not necessary for the story, or if they were necessary then they were not sufficiently described in the text for me to be able to understand. In either case, I want to assert that this story is so strong that it does not need any extra data and as mentioned above in the section about the acetylation differences at enhancer and promoters, including too many other less important analyses may well be hiding data that supports the main points.

We agree with the Reviewer in this point. We were very excited with the novel insight and answers provided by our study. We initially submitted the article to *Nature*, *Nature Neuroscience* and *Nature Cell Biology*, but the editors handling the manuscript decided to reject the article without peer review. In the time between submissions (a few months), we conducted new analyses with the hope to surpass the editorial filter. Maybe a downside effect of this effort is that the article is less focused. Following the advise of the Reviewer we eliminated some Supplementary panels in Fig. S11 (former Fig. S10) that did not contribute to the main message of our study (e.g., IHC questioned by Reviewer #2 and the Hi-C analyses referred by Reviewer #1). For the same reason, we decided to move some panels for Fig. 6 to Supplemental Fig. S13. The elimination of these accessory panels allowed the presentation of the new analysis requested by the Reviewers without a larger increase in the number of Figures or exceeding the recommended number of characters in the main text.

Reviewer #2 (Remarks to the Author):

1- General comments

The manuscript “Neuronal identity is maintained in the adult brain through KAT3-dependent enhancer acetylation” by Lipinski and colleagues explores the role of two key acetyltransferases, (CBP and p300) in cell-fate maintenance. Using inducible knock-out animals, they show that the ablation of both genes leads to loss of specific neuronal identity, exemplified by the loss of electrical activity, proper connections and aberrant dendrite morphologies. The authors describe in genome-wide assays the consequence of the loss of function. Using a combination of transcriptomics and chromatin accessibility approaches, they document the specificity of action of CBP and p300, and propose some of the mechanisms by which these chromatin remodelers can lock specific neuronal populations and astrocytes in their respective identities.

Overall this manuscript is a well-written documentation combining in vivo and in vitro experimental series to demonstrate the role of KAT3 in the adult brain. It extends the field by detailing the specificity of acetylation marks on neuronal genes to maintain cell-fate. Altogether their findings are convincing and the questions being addressed are likely to further influence the way we will develop ideas in the field. Furthermore, the experimental design of the study is robust and in general the data appears technically rigorous.

We also thank Reviewer #2 for the positive comments and summary of our main findings.

2- Specific comments are listed below:

-The central point that is made by the authors regards the loss of neuronal identity. While it is clear that neuronal function is altered (activity, synapses, dendritic morphologies), it is still uncertain how non-neuronal the cells become. The conserved proportion of NeuN positive cells in the hippocampi from dKAT3-fKO suggest that the neuronal state is still maintained. Also, the conclusion that they acquire a “molecularly undefined fate” sounds elusive. It would be informative to use the transcriptomic data to define more precisely define the default cell type or cell state that the mutant cells adopt. Stating that they do not express any markers is not sufficient. At least the authors should report more clearly in the text or the

main figures the main genes that are specifically expressed in these so-called non-neuronal population.

We believe that our data clearly demonstrate major changes in the cellular transcriptome and epigenome driven by the loss of CBP and p300. These changes seem to primarily affect neuronal-specific genes. The genomic changes correlate with dramatic changes in neuronal morphology and physiology. We summarize all these events as a loss of neuronal identity, but acknowledge the difficulty of defining “cellular identity”; the term itself is currently evolving and the rapid advance in single-cell transcriptome and epigenome analysis has led to an almost philosophical debate distinguishing between “identity” and “state”. We do not aim to solve this debate here, but to underscore the particular relevance of the paralog pair CBP/p300 in the maintenance of cell type specific gene programs. Regarding some of the specific points raised by the Reviewer:

- *“The conserved proportion of NeuN positive cells in the hippocampi from dKAT3-ifKO suggest that the neuronal state is still maintained”.* The impact of the combined CBP and p300 ablation in hippocampal neurons is very homogeneous (the very few cells in the cellular layers of the hippocampus that still express NeuN are likely interneurons). Therefore, we do not understand what the Reviewer meant by “conserved proportion”. The reduction in NeuN expression is presented in several figures. More precisely in Figure 2e (in which the dKAT3-ifKO mouse was perfused 3 weeks after the last TMX injection), 2f (in which the animal was perfused 5 weeks after viral infection with the AVV-cre), and S4e (in which the dKAT3-ifKO mouse was perfused 3 weeks after the last TMX injection). In all the cases, the reduction in NeuN immunoreactivity is very dramatic. Particularly, when there was a larger delay between gene recombination and perfusion. Our genetic manipulation causes the disruption of the *Crebbp* and *Ep300* genes, this leads first to a loss of CBP and p300 functional transcripts, then to a loss of CBP and p300 immunoreactivity and lysine hypoacetylation that likely underlies the transcriptional shutdown observed in neuronal genes. Once the transcript levels of downstream genes are down, the decrease in the levels of proteins will follow, but the detection of protein elimination would require some time depending on the specific stabilities of the downstream protein target. The loss of identity is a progressive process detectable at many different levels. We present in the attached **Figure 2 for Reviewers** some additional images demonstrating the loss of NeuN expression in dKAT3 neurons.

- *“It would be informative to use the transcriptomic data to define the default cell type or cell state that the mutant cells adopt”.* We used the wealth of genomic data generated in our study and conducted a number of comparative analyses with other recent studies in this field. Remarkably, our fully unbiased analyses show that the cell cluster markers retrieved in our snRNA-seq experiment or in other scRNA-seq screens are not longer expressed in dKAT3-KO cells. As shown in Figure 3e, the cKO-specific clusters are not identified for the expression of any specific gene but for the lack of expression of the genes that enabled the classification of the other clusters. This conclusion is further strengthened with the new trajectory analysis included in Figure 3f-g that is discussed below.

- *“...the authors should report more clearly ... the main genes that are specifically expressed in these so-called non-neuronal population”.* These genes were listed in Supplementary Table S3. This is now more clearly indicated in the text.

-While the text is well written, wording is often vague. I understand but it would be easier to grasp the details with a more precise description of the results in the text (See eg. l. 114, 115, 120, 121, 123, 127, 128, 150, 193, 258...).

We thank the Reviewer for pointing out these sentences. We revised the text to be more precise in our wording.

- The use of an inducible system to ablate specifically the KAT proteins in the forebrain is a suitable way to address the question. However, the Cre used here (Camk2a-Cre/ERT2) has been reported to already demonstrate activity even in the absence of tamoxifen (Madisen et al., 2010; doi:10.1038/nn.2467). Did the authors take into account the sparse expression of the Cre prior the induction? Would it be possible to mention it in the text or discussion?

The Reviewer should note that the strain used here is not the one used by Madisen and colleagues, but the one generated by the lab of Gunter Schutz (DKFZ, Heidelberg, Germany). The Cre line analyzed by Madisen and colleagues was generated using a short (1.3 kb) Camk2a promoter and drives 'leaky' expression in scattered excitatory neuronal populations in the non-induced state (according to the authors). Instead, the line generated by Erdmann, Schutz and Berger was produced using a BAC vector harboring a genomic insert containing the CaMKII α gene locus with a 43 kb 5'upstream and a 100 kb 3'downstream region and shows minimal leakiness in the adult brain (Erdmann et al., 2007). We have extensive experience using this particular Camk2a-CreERT strain and did not detect significant recombination in brain cells before TMX treatment. This is particularly evident after crossing this line with the strong reporter for cre-driver lines CMV-fxSTOP-tdTomato (we include some images for Reviewer #2 perusal in Figure 3 for Reviewers).

-The description of the phenotype (second paragraph of the results section, l.117-122) indicates that there is a failure of the mice to hydrate/feed properly. How much of the hydration problem still persists when the animals obtained facilitated access to food/water? Have you performed any test to control for the restoration of proper hydration and weight?

We found that the mice gained some weight in the first week after gene ablation but they later rapidly loss weight. The facilitation of the access to water and food slowed down this process but did not stop it. We have not specifically investigated hydration because this phenomenon is outside the main scope of our study.

-Figure 1d-e: How confident can we be in the phenotype observed when the t-test is performed with n of 2 in the mutant condition (Fig. 1d)? What about the ANOVA (Fig. 1e) with two small groups as well (n=2 after 2 months, n=1 after 7 months). These low "n" values may somewhat impede the power of the analysis of these important phenotypes. How assertively can we interpret these results?

As indicated in the text, most mice do not reach that time after gene ablation because they either died or were sacrificed before they reached that point. Increasing the n for this conditions will require an enormous number of mice treated with TMX (note that more than 90% will die in the first weeks) and will be again the 3R in animal research because it will not provide additional insight and will increase the suffering inflicted to the mice. We used proper statistical test to deal with the reduced "n". We believe that we are not particularly "assertive" with the interpretation of the results (which are secondary in the context of the whole study), but we could eliminate these analyses if the Reviewer advises so.

-The OB in the mutant looks very large. Is that often observed? Is the Cre expressed here as well?

We did not appreciate any consistent increase in the size of the olfactory bulb (see also the response to the previous point). The creERT2 recombinase is not expressed in this area.

- The single-cell transcriptomes are shown using a UMAP, which give a good overview of the

molecular diversity. It also reveals an apparent regression in cell-fate specification. However, since the trend indicates a loss of differentiation, to better emphasize the shift toward undefined cell states I would recommend a trajectory-based representation/organization. Organizing the cells across a pseudotemporal alignment may help discriminate between a change in cell type and a regression along a differentiation axis.

We thank the Reviewer for this suggestion. Conversely to other dimensionality reduction methods, such as tSNE (van der Maaten and Hinton, 2008, *Journal of Machine Learning Research*), UMAP tends to better preserve the global structure of the data in the low dimensional space (McInnes et al., arXiv:1802.03426). Nevertheless, since the cluster distances of the manifolds cannot be directly interpreted in this type of representation, we complemented our initial analysis with a pseudo-temporal trajectory analysis using DDRTree (Qi et al., 2015) as suggested by the Reviewer. These analyses are presented in the new Figures 3f and 3g and further strengthen our conclusions regarding an undifferentiated cell state.

-The authors report a slight change in heterochromatin structures (Fig S3c-d). How does this phenotype relate to the absence of H3K9me3 levels reported? (See Result section; l. 271-273)

Our results indicate that there is no prominent change in H3K9me3 levels but possibly a reorganization of genetic material in the nucleus interior. Following the advice of Reviewer #1, we eliminated the immunostaining against H3K9me3 because this result seems distracting and does not relate to the main findings described in our manuscript. We hope that Reviewer #2 will agree with this decision.

-The rescue experiment is a very interesting attempt to further explore the mechanistic specificity of the KAT phenotypes. This rescue being conducted on Neurod2 based on a severe downregulation, how does this downregulation compare to the effects on other genes? Is there any other reason to select that specific TF? It would have been helpful to evaluate how negative control genes behave upon such CRISPR modulation.

These experiments are very challenging. We attempted to rescue expression at other loci with less success. The great quality and sensitivity of the available antibody for Neurod2 and the nuclear localization of this antigen facilitated the assessment of the rescue in the case of Neurod2. The Reviewer should note that Figures 8c-d and 8f-g already presented the requested negative control. As indicated in the legend, a gRNA targeting the hippocalcin promoter was used as a control for the specificity of the effect.

-It is not clear to me how the one day between “gRNA Cherry transfection + dCas9KAT” and the analysis is enough to elicit the described effect.

We do not understand this comment. We introduced the constructs encoding dCas9KAT and the gRNA and observed the enhanced transcription of the targeted locus. The result suggests a rapid and direct effect.

- In describing the Movie M1 the authors state that “Shortly afterwards, the same mice showed severe ataxia, and loss of the righting reflex, escaping response and tail-suspension-evoked stretching”. However, there is no clear quantification of these effects. The behavior there could be explained by sleep or food disorder, or by a sporadic sickness.

We agree with the Reviewer: the behavior there could be explained by sleep or food disorder, sporadic sickness and other discomforts. We really do not know how the massive loss of neuronal identity in broad brain regions (cortex, amygdala, hippocampus and other essential brain nuclei) may interfere with all these epiphenomena.

3- Minor points:

-The first words of the abstract tend to diminish the work from other groups. “Very” is not necessary.

We eliminated the word “very” in the referred sentence.

-The study is based on experiments mostly in hippocampal neurons and astrocytes. This should be explicitly stated in the abstract.

We added this information in the abstract.

-Line 104: “while brain regions in which the CaMKII α promoter is not active, such as the cerebellum and the basal ganglia, were spared.” Could you add a reference to this statement?

This is shown in the original paper describing the mice (Erdmann et al. 2007). We added this reference to the statement (this reference was included in the list of Supplementary references but is has been now moved to the main References list). The pattern has been corroborated in numerous articles using this strain (including several studies conducted in our lab). We also present evidence supporting this statement in Figure S1.

-Line 119: Is the data related to the statement shown? (“all mice died within the first 2-3 weeks after TMX administration”)

No. This sentence refers exclusively to the first litters. Later on, as described in the text, we improved the survival of the mice. The sentence has been clarified and we thank the Reviewer for pointing this out.

-Line 260: How enhancers were defined? A reference? By the presence of p300? Chromatin landmarks?

This was explained in the Experimental procedures section (p. 16): “We defined as neuronal enhancers those regions that contain neuronal peaks for KAT3 binding, ATAC-seq (i.e., regions that are occupied), H3K27ac, enriched in H3K4me1 and located in introns or intergenic regions (beyond 1 Kb of the TSS)”. We have added a recent reference to this definition. When comparing to other enhancer definitions in the literature, we believe that we are using a particularly stringent definition the number of enhancer will be higher if any of these criteria is eliminated.

-Fig. 8c and 8f: The scale bars are missing.

We added the missing scale bars. We thank the Reviewer for noticing their absence.

-Fig S3c: What was the statistical test performed here?

These are unpaired t-tests. The information was added to the legend.

-Fig. S3g: any quantification for the Cas3 signals?

We did not detect dCas3 positive cells in either system. The comparison of the percentages (null in all the cases) indicated a non-significant difference.

-Fig. S6c: “staining against ...genes NeuroD2”: is the gene stained or is it the protein? Why genes is plural?

We corrected this sentence. We apologize for the mistake.

-Fig. S7: In the caption title it is RNA-seq (not RNA-seg).

We have corrected the caption title.

-Fig. S10c: H3K9me3 seems reduced in the image shown. Same in panel S10a. Did the authors measure/quantify the levels of this histone mark?

No, we did not measure H3K9me3 levels in immunohistochemistry images. This technique is not very quantitative, particularly when the signal is so irregularly distributed as in the case of H3K9me3. We only intended to present a qualitative assessment. As indicated above, we have decided to eliminate this result because it is not directly related to the main findings and message of our study.

-Fig. S10g: do you mean one-tailed Mann-Whitney test?

Yes. We corrected the text in the figure legend.

Figure 1 for Reviewers. Neuronal nuclei size and ultrastructure in dKAT3-*if*KOs one month after TMX treatment. a. Confocal image of CA1 pyramidal neuron nuclei stained with DAPI in dKAT3-*if*KO and control mice. **b.** Low magnification electron microscopy image showing the cellular layer and *stratum radiatum* in a dKAT3-*if*KO and a control littermate. **c.** Electron microscopy images of CA1 pyramidal neuron nuclei from dKAT3-*if*KOs and a control littermates. **d.** Confocal image of the nuclei of granule cells in the dentate gyrus stained with DAPI in dKAT3-*if*KO and control mice. **e.** Electron microscopy images of granule cell nuclei from dKAT3-*if*KOs and a control littermates.

Figure 2 for Reviewers. The combined ablation of CBP and p300 causes a dramatic and consistent reduction of NeuN immunoreactivity. a. This panel presents the same result than Figure 2e in a different mouse. **b.** This panel presents the same result than Figure 2f but provides a wider field and demonstrates the specific loss of NeuN immunoreactivity in dentate gyrus neurons (infected with the cre-expressing AVV, in green) compared to CA1 pyramidal neurons (not infected with the AVV). Note the loss of NeuN immunoreactivity in the dentate gyrus, CA3 and subicullum, the three areas that show AAV infection.

Figure 3 for Reviewers. Expression of a recombination reporter gene (tdTomato) in in the brain of CamKII-creERT2 mice after treatment with TMX. a. Brain of control mice not treated with TMX. **b.** TdTomato fluorescence one week after TMX treatment. **c.** TdTomato fluorescence two months after TMX treatment.

Reviewers' Comments:

Reviewer #1:

Remarks to the Author:

The authors have addressed all of my concerns.

Reviewer #2:

Remarks to the Author:

The revision of the manuscript by Lipinski and colleagues provides detailed point-by-point answers to the comments and concerns raised initially by the reviewers. The experiments are properly designed and analyzed, and the data are convincing. The level of details in the methods and the description provided appear sufficient to ensure reproducibility. This is an important work that will significantly expand our understanding of chromatin regulation in neuronal fate maintenance and will be of great interest for the broad readership of Nature Communications. Overall, the revisions are satisfactory.

Please find below the answers (>) to the point-by-point list given by the authors (*):

*"The conserved proportion of NeuN positive....the reduction in NeuN immunoreactivity is very dramatic. "

> It is clear that the ablation of CBP/EP300 does reduce the expression of neuronal-specific genes, including NeuN. While the downregulation of Rbfox3 is indeed robust and homogenous, the levels of the messenger (Fig. 2i) and the protein (Fig. 2e) appear to be both present/positive (even at lower levels), a hallmark of neuronal fate. Thus, the claim "loss of neuronal identity" appears more as "reduction of neuronal-specific gene expression".

* "Particularly, when there was a larger delay ...expression in dKAT3 neurons"

>This is convincing.

* "It would be informative to use the transcriptomic ...Figure 3f-g that is discussed below"

> This new analysis in Fig 3f-g is interesting and helps clarify the different levels of differentiation. To support the claim of the loss of neuronal identity, a suggestion would be to aggregate the transcriptomes, including the mutant cells, with a variety of tissue and perform hierarchical clustering to see how these iKO would segregate from the neurons (or not).

* "These genes were listed in Supplementary Table S3. This is now more clearly indicated in the text."

> The mention in the text does better refer to the Table S3. In this table however, the genes listed as top 10 markers (shown in the first tab) of the excel file are ranked by p-value, so that in the two "former" neuronal cells most of the genes are actually depleted (negative logFC). While this information about the depleted genes is important (e.g. Cux2, Mef2c, Zeb2...in former CA_pyr), it does not reveal what define their distinctive signature.

* "The Reviewer should note that the strain used ..."

>Thanks for adding this information.

* "We have extensive experience"

> Thanks.

* "We found that the mice gained some weight..."

> Thanks for the explanation.

* "We have not specifically investigated hydration..."

> Could it be a causal nexus with the dehydration as an intermediate step responsible for the shrinkage of affected neurons? However, I agree it is beyond the central message of this study.

* "As indicated in the text, most mice do not reach that time after gene ablation.."

> Yes this information should stay. It is an interesting observation as long as the statistics are rigorous.

* "We did not appreciate any consistent increase in the size of the olfactory bulb (see also the response to the previous point). The creERT2 recombinase is not expressed in this area."

> OK.

* "We thank the Reviewer for this suggestion. Conversely to other dimensionality..."

> Thanks for performing this additional analysis. The rendering looks good. Perhaps an arrow pointing toward the undifferentiated state would be informative here.

* "Our results indicate that there is no prominent change in H3K9me3...We hope that Reviewer #2 will agree with this decision."

> While showing H3K9me3 maintenance in the mutant may not be necessary, it is still an important information, especially when the heterochromatin is modified.

* "These experiments are very challenging. We attempted to rescue expression at other loci with less success."

> That was the question, how much Neurod2 is selected, and how it compares to other genes that are less prone to rescue the phenotype.

* "The great quality...as a control for the specificity of the effect"

> I agree that these experiments are challenging and the point made here is reasonable.

* "We do not understand this comment. We introduced the constructs encoding dCas9KAT and the gRNA and observed the enhanced transcription of the targeted locus. The result suggests a rapid and direct effect."

> The comment could be read as: a longer time (48-72h) between transfection and analysis may have given more robust results.

*3- Minor points:

>>The response to all these minor points is satisfactory.

Pierre Fabre

Point-by-point response to Reviewers

We thank the Editor and Reviewers for their comments. We have revised the text and amended Figure 3 and Supplementary Figure 11 according to Reviewer#2 suggestions. Our point-by-point response to the last concerns expressed by Reviewer#2 is below.

Reviewer #1 (Remarks to the Author):

The authors have addressed all of my concerns.

We thank again the Reviewer#1 for his/her very useful comments that helped us to improve our manuscript.

Reviewer #2 (Remarks to the Author):

The revision of the manuscript by Lipinski and colleagues provides detailed point-by-point answers to the comments and concerns raised initially by the reviewers. The experiments are properly designed and analyzed, and the data are convincing. The level of details in the methods and the description provided appear sufficient to ensure reproducibility. This is an important work that will significantly expand our understanding of chromatin regulation in neuronal fate maintenance and will be of great interest for the broad readership of Nature Communications. Overall, the revisions are satisfactory.

Please find below the answers (>) to the point-by-point list given by the authors ():*

We thank the Reviewer#2 for the positive appreciation of our work and thoughtful revision. We respond below each of his additional comments.

**“The conserved proportion of NeuN positive.... the reduction in NeuN immunoreactivity is very dramatic.”*

> It is clear that the ablation of CBP/EP300 does reduce the expression of neuronal-specific genes, including NeuN. While the downregulation of Rbfox3 is indeed robust and homogenous, the levels of the messenger (Fig. 2i) and the protein (Fig. 2e) appear to be both present/positive (even at lower levels), a hallmark of neuronal fate. Thus, the claim “loss of neuronal identity” appears more as “reduction of neuronal-specific gene expression”.

We thank the reviewer for this comment.

The description of Fig. 2e (lines 183-84) indicates “Immunodetection experiments for neuronal proteins like CaMKIV, NeuN and hippocalcin confirmed the dramatic loss of expression of neuronal proteins” (very much as suggested by the Reviewer).

The description of Fig. 2i (lines 197-98) indicates “They do, however, show... the specific downregulation of neuron-specific transcripts” (also as suggested by the Reviewer).

The general conclusion of “loss of neuronal identity” is not based on the images of NeuN immunostaining or on the traces of RNA-seq for that particular gene (*Rbfox3*). This conclusion arises from the integration of all the functional assays and genomic analyses conducted in the paper. Therefore, we do not believe that any correction in the text is needed regarding this statement.

** "Particularly, when there was a larger delay ...expression in dKAT3 neurons"*

>This is convincing.

Thanks.

** “It would be informative to use the transcriptomic ...Figure 3f-g that is discussed below”*

> This new analysis in Fig 3f-g is interesting and helps clarify the different levels of differentiation. To support the claim of the loss of neuronal identity, a suggestion would be to

aggregate the transcriptomes, including the mutant cells, with a variety of tissue and perform hierarchical clustering to see how these iKO would segregate from the neurons (or not). We are sorry but we do not understand which is the specific additional analysis requested by the Reviewer (in particular, we do not understand what *variety of tissues* is he referring to). We are also concerned with the suitability and utility of adding a hierarchical clustering representation to our comprehensive analysis of the snRNA-seq experiment (we already dedicate three full figures with dozens of graphs to the description of this experiment). Moreover, we performed unsupervised clustering using two different graph-based algorithms: shared nearest neighbor (SNN) with Louvain modularity optimization (Stuart et al., 2019), and reverse graph embedding (Qiu et al., 2017). Importantly, both algorithms consistently clustered dKAT3-KO cells apart from their original putative neuronal populations (Fig. 3d-g). Notably, our longitudinal experimental design revealed a temporal sequence of cellular events leading to progressive loss of neuronal identity (Fig. 3d-g). We observed a progressive loss of gene markers of canonical excitatory neuronal classes in the hippocampus and markers of excitatory neurons (Supplementary Fig. 7). These findings add to many other results throughout the study and together, support that CBP/p300 play a central role in the maintenance of specific gene expression programs underlying neuronal identity. We prefer not to overload the article with additional analysis that will not reveal any novel insight. See also our related response below.

** "These genes were listed in Supplementary TableS3. This is now more clearly indicated in the text."*

*> The mention in the text does better refer to the Table S3. In this table however, the genes listed as top 10 markers (shown in the first tab) of the excel file are ranked by p-value, so that in the two "former" neuronal cells most of the genes are actually depleted (negative logFC). While this information about the depleted genes is important (e.g. Cux2, Mef2c, Zeb2...in former CA_pyr), it does not reveal what define their distinctive signature. Indeed, this is one of the most interesting findings of our snRNA-seq experiment. **There are no genes specifically overexpressed in these cells when compared to the other cells.** The genes that are overexpressed in "former" neuronal cells do not meet the parameters required to fall in the category of "marker genes". First, these genes do not show selective expression in "former" neurons (this can be checked by comparing pct.1 (test cluster) vs pct.2 (reference, all the other cells) in Table S3). Second, conversely to canonical marker genes of known cell types, upregulated genes in "former" neurons show mild enrichment levels (i.e., never reached values of average log fold change > 0.7; see Supplementary Table 3). Strikingly, the most differentially expressed genes in these cells are downregulated and are genes related to neuronal function (which is consistent with the bulk RNA-seq experiment). All this information can be easily retrieved by exploring the different sheets of the excel file in Supplementary Table 3. Maybe the Reviewer focused on the first tab and overlooked the information presented in the other twelve sheets.*

** "The Reviewer should note that the strain used ..."*

>Thanks for adding this information.

** "We have extensive experience"*

> Thanks.

** "We found that the mice gained some weight..."*

> Thanks for the explanation.

* *"We have not specifically investigated hydration..."*

> *Could it be a causal nexus with the dehydration as an intermediate step responsible for the shrinkage of affected neurons? However, I agree it is beyond the central message of this study.*

* *"As indicated in the text, most mice do not reach that time after gene ablation.."*

> *Yes this information should stay. It is an interesting observation as long as the statistics are rigorous.*

* *"We did not appreciate any consistent increase in the size of the olfactory bulb (see also the response to the previous point). The creERT2 recombinase is not expressed in this area."*

> *OK.*

We are glad that the Reviewer found our response to these points satisfactory.

* *"We thank the Reviewer for this suggestion. Conversely to other dimensionality..."*

> *Thanks for performing this additional analysis. The rendering looks good. Perhaps an arrow pointing toward the undifferentiated state would be informative here.*

We thank the Reviewer for this suggestion. We added the requested arrows to Figure 3f and 3g.

* *"Our results indicate that there is no prominent change in H3K9me3...We hope that Reviewer #2 will agree with this decision."*

> *While showing H3K9me3 maintenance in the mutant may not be necessary, it is still an important information, especially when the heterochromatin is modified.*

Following the Reviewer's advise we added back panel 11c to Supplementary Figure 11. As indicated, in our respond, we only intended to make a qualitative assessment because we consider that the quantification of the signal will not be very reliable.

* *"These experiments are very challenging. We attempted to rescue expression at other loci with less success."*

> *That was the question, how much Neurod2 is selected, and how it compares to other genes that are less prone to rescue the phenotype.*

* *"The great quality...as a control for the specificity of the effect"*

> *I agree that these experiments are challenging and the point made here is reasonable.*

We are glad that the Reviewer found our response to these points satisfactory.

The *Neurod2* locus was selected for different reasons: (i) *Neurod2* is one of top downregulated genes in dKAT3-ifKOs; (ii) the locus also show a drastic reduction on H3K27ac; (iii) among the antibodies that we had available, the anti-ND2 antibody worked very well which greatly facilitated the analysis. Also we had conducted the NeuroD2 overexpression experiments (Supplementary Fig. 14) showing that bHLH alone cannot recover the phenotype, which led to interesting conclusions.

The gRNAs design for *Neurod2* was targeted to an ATAC-seq peak specific of pyramidal neurons that overlapped with CBP&p300 peaks that disappear in dKAT3-ifKOs. However, the ATAC-seq peak does not disappear in dKAT3-ifKOs, indicating that the promoter is still occupied by other proteins. As can be see in the snapshots presented in Figure 5l and 8a (dark gray traces), the ATAC-seq profile is much more affected in *Hpca* (which could not be rescued) than in *Neurod2* (which was rescued). This difference can contribute to explain the

distinct output of the epi-editing experiments. However, we also examined *Bdnf* that presents a profile more similar to *Neurod2* and was apparently not rescued. This result is not included in our manuscript because we consider that this negative result was not very reliable given the low quality of the BDNF antibody and its diffuse signal in soma and dendrites, particularly when compared to the salient signal for ND2 that concentrates in the neuronal nuclei. We have now included a sentence in page 18 discussing this point, but we prefer not to make a general statement about the accessibility pattern and the rescue of the locus based only in 2 or 3 examples.

** "We do not understand this comment. We introduced the constructs encoding dCas9KAT and the gRNA and observed the enhanced transcription of the targeted locus. The result suggests a rapid and direct effect."*

> The comment could be read as: a longer time (48-72h) between transfection and analysis may have given more robust results.

We understand now the question. We chose to focus our analysis 24 h after transfection. Neuronal cells suffer after transfection and we did not want to expose the cell to additional stress. Since our first experiments revealed a clear upregulation of NeuroD2 after 24 h, subsequent experiments used the same time point.

**3- Minor points:*

>>The response to all these minor points is satisfactory.

We are glad that the Reviewer found our response to these points satisfactory.

Reviewers' Comments:

Reviewer #2:

Remarks to the Author:

Thanks for the answers to my last comments.

I have no more concerns. Concerning the point about the snRNA-seq (and the Table S3), thanks for making things clear. It took a long time reviewing the twelve sheets, which is why I proposed to clarify the first tab of the Table S3. Thus, I agree this information can be retrieved, but nevertheless it would be more helpful to read the differentially expressed genes listed by fold change enrichment (even if there is not much upregulation, the one that are upregulated can be informative (e.g. *Onecut2*, see doi: [10.1093/nar/gkz273](https://doi.org/10.1093/nar/gkz273)))

Best regards

Point-by-point response to Reviewer #2 and editorial requests

We thank the Editor and Reviewer #2 for their comments. We have revised our paper according to these last instructions. In addition, I would like to inform you the mRNA-seq, ATAC-seq, ChIP-seq, snRNA-seq datasets generated in this study are publicly available with the GEO accession number GSE133018.

Reviewer #2 (Remarks to the Author):

Thanks for the answers to my last comments. I have no more concerns. Concerning the point about the snRNA-seq (and the Table S3), thanks for making things clear. It took a long time reviewing the twelve sheets, which is why I proposed to clarify the first tab of the Table S3. Thus, I agree this information can be retrieved, but nevertheless it would be more helpful to read the differentially expressed genes listed by fold change enrichment (even if there is not much upregulation, the one that are upregulated can be informative (e.g. Onecut2, see doi: 10.1093/nar/gkz273))

We have revised the Supplementary Table 3. We added a new tab that presents the information requested by Reviewer#2 (Top enriched genes). In addition, we have revised the 12 spreadsheets corresponding to each cluster to facilitate the identification of genes displaying a significant enrichment, significant depletion or not significant change. The genes remain ordered by significance because we believe that this better captures population-level effects as compared to fold change enrichment.